# Parametrized Quantum Policies
# for Reinforcement Learning

**Sofiene Jerbi**
Institute for Theoretical Physics,
University of Innsbruck
`sofiene.jerbi@uibk.ac.at`

**Casper Gyurik**
LIACS,
Leiden University

**Simon C. Marshall**
LIACS,
Leiden University

**Hans J. Briegel**
Institute for Theoretical Physics,
University of Innsbruck

**Vedran Dunjko**
LIACS,
Leiden University

## Abstract

With the advent of real-world quantum computing, the idea that parametrized quantum computations can be used as hypothesis families in a quantum-classical machine learning system is gaining increasing traction. Such hybrid systems have already shown the potential to tackle real-world tasks in supervised and generative learning, and recent works have established their provable advantages in special artificial tasks. Yet, in the case of reinforcement learning, which is arguably most challenging and where learning boosts would be extremely valuable, no proposal has been successful in solving even standard benchmarking tasks, nor in showing a theoretical learning advantage over classical algorithms. In this work, we achieve both. We propose a hybrid quantum-classical reinforcement learning model using very few qubits, which we show can be effectively trained to solve several standard benchmarking environments. Moreover, we demonstrate, and formally prove, the ability of parametrized quantum circuits to solve certain learning tasks that are intractable to classical models, including current state-of-art deep neural networks, under the widely-believed classical hardness of the discrete logarithm problem.

## 1    Introduction

Hybrid quantum machine learning models constitute one of the most promising applications of near-term quantum computers [1, 2]. In these models, parametrized and data-dependent quantum computations define a hypothesis family for a given learning task, and a classical optimization algorithm is used to train them. For instance, parametrized quantum circuits (PQCs) [3] have already proven successful in classification [4–8], generative modeling [9, 10] and clustering [11] problems. Moreover, recent results have shown proofs of their learning advantages in artificially constructed tasks [6, 12], some of which are based on widely believed complexity-theoretic assumptions [12–15]. All these results, however, only consider supervised and generative learning settings.

Arguably, the largest impact quantum computing can have is by providing enhancements to the hardest learning problems. From this perspective, reinforcement learning (RL) stands out as a field that can greatly benefit from a powerful hypothesis family. This is showcased by the boost in learning performance that deep neural networks (DNNs) have provided to RL [16], which enabled systems like AlphaGo [17], among other achievements [18, 19]. Nonetheless, the true potential of near-term quantum approaches in RL remains very little explored. The few existing works [20–23] have failed so far at solving classical benchmarking tasks using PQCs and left open the question of their ability to provide a learning advantage.

35th Conference on Neural Information Processing Systems (NeurIPS 2021).

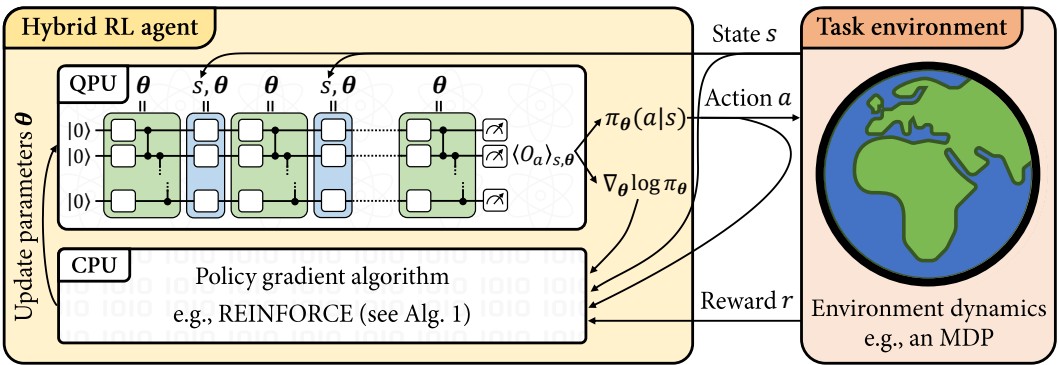

Figure 1: **Training parametrized quantum policies for reinforcement learning.** We consider a quantum-enhanced RL scenario where a hybrid quantum-classical agent learns by interacting with a classical environment. For each state $s$ it perceives, the agent samples its next action $a$ from its policy $\pi_{\boldsymbol{\theta}}(a|s)$ and perceives feedback on its behavior in the form of a reward $r$. For our hybrid agents, the policy $\pi_{\boldsymbol{\theta}}$ is specified by a PQC (see Def. 1) evaluated (along with the gradient $\nabla_{\boldsymbol{\theta}} \log \pi_{\boldsymbol{\theta}}$) on a quantum processing unit (QPU). The training of this policy is performed by a classical learning algorithm, such as the REINFORCE algorithm (see Alg. 1), which uses sample interactions and policy gradients to update the policy parameters $\boldsymbol{\theta}$.

**Contributions**   In this work, we demonstrate the potential of policies based on PQCs in solving classical RL environments. To do this, we first propose new model constructions, describe their learning algorithms, and show numerically the influence of design choices on their learning performance. In our numerical investigation, we consider benchmarking environments from OpenAI Gym [24], for which good and simple DNN policies are known, and in which we demonstrate that PQC policies can achieve comparable performance. Second, inspired by the classification task of Havlíček *et al.* [6], conjectured to be classically hard by the authors, we construct analogous RL environments where we show an empirical learning advantage of our PQC policies over standard DNN policies used in deep RL. In the same direction, we construct RL environments with a provable gap in performance between a family of PQC policies and any efficient classical learner. These environments essentially build upon the work of Liu *et al.* [14] by embedding into a learning setting the discrete logarithm problem (DLP), which is the problem solved by Shor's celebrated quantum algorithm [25] but widely believed to be classically hard to solve [26].

**Related work**   Recently, a few works have been exploring hybrid quantum approaches for RL. Among these, Refs. [20, 21] also trained PQC-based agents in classical RL environments. However, these take a value-based approach to RL, meaning that they use PQCs as value-function approximators instead of direct policies. The learning agents in these works are also tested on OpenAI Gym environments (namely, a modified FrozenLake and CartPole), but do not achieve sufficiently good performance to be solving them, according to the Gym specifications. Ref. [27] shows that, using some of our design choices for PQCs in RL (i.e., data re-uploading circuits [28] with trainable observable weights and input scaling parameters), one can also solve these environments using a value-based approach. An actor-critic approach to QRL was introduced in Ref. [22], using both a PQC actor (or policy) and a PQC critic (or value-function approximator). In contrast to our work, these are trained in quantum environments (e.g., quantum-control environments), that provide a quantum state to the agent, which acts back with a continuous classical action. These aspects make it a very different learning setting to ours. Ref. [23] also describes a hybrid quantum-classical algorithm for value-based RL. The function-approximation models on which this algorithm is applied are however not PQCs but energy-based neural networks (e.g., deep and quantum Boltzmann machines). Finally, our work provides an alternative approach to take advantage of quantum effects in designing QRL agents compared to earlier approaches [29–33], which are mainly based on (variations of) Grover's search algorithm [34] or quantum annealers [35] to speed up sampling routines.

**Code**   An accompanying tutorial [36], implemented as part of the quantum machine learning library TensorFlow Quantum [37], provides the code required to reproduce our numerical results and explore different settings. It also implements the Q-learning approach for PQC-based RL of Skolik *et al.* [27]

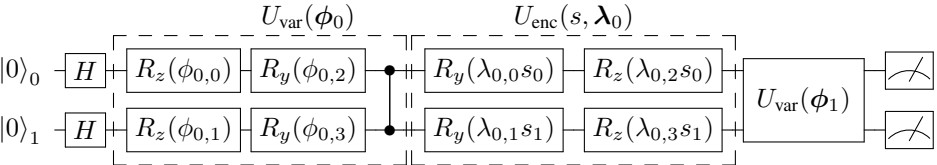

Figure 2: **PQC architecture for $n = 2$ qubits and depth $D_{\text{enc}} = 1$.** This architecture is composed of alternating layers of encoding unitaries $U_{\text{enc}}(s, \boldsymbol{\lambda}_i)$ taking as input a state vector $s = (s_0, \ldots, s_{d-1})$ and scaling parameters $\boldsymbol{\lambda}_i$ (part of a vector $\boldsymbol{\lambda} \in \mathbb{R}^{|\boldsymbol{\lambda}|}$ of dimension $|\boldsymbol{\lambda}|$), and variational unitaries $U_{\text{var}}(\boldsymbol{\phi}_i)$ taking as input rotation angles $\boldsymbol{\phi}_i$ (part of a vector $\boldsymbol{\phi} \in [0, 2\pi]^{|\boldsymbol{\phi}|}$ of dimension $|\boldsymbol{\phi}|$).

## 2 Parametrized quantum policies: definitions and learning algorithm

In this section, we give a detailed construction of our parametrized quantum policies and describe their associated training algorithms. We start however with a short introduction to the basic concepts of quantum computation, introduced in more detail in [38, 39].

### 2.1 Quantum computation: a primer

A quantum system composed of $n$ qubits is represented by a $2^n$-dimensional complex Hilbert space $\mathcal{H} = (\mathbb{C}^2)^{\otimes n}$. Its quantum state is described by a vector $|\psi\rangle \in \mathcal{H}$ of unit norm $\langle\psi|\psi\rangle = 1$, where we adopt the bra-ket notation to describe vectors $|\psi\rangle$, their conjugate transpose $\langle\psi|$ and inner-products $\langle\psi|\psi'\rangle$ in $\mathcal{H}$. Single-qubit computational basis states are given by $|0\rangle = (1, 0)^T, |1\rangle = (0, 1)^T$, and their tensor products describe general computational basis states, e.g., $|10\rangle = |1\rangle \otimes |0\rangle = (0, 0, 1, 0)$.

A quantum gate is a unitary operation $U$ acting on $\mathcal{H}$. When a gate $U$ acts non-trivially only on a subset $S \subseteq [n]$ of qubits, we identify it to the operation $U \otimes \mathbb{1}_{[n] \setminus S}$. In this work, we are mainly interested in the single-qubit Pauli gates $Z, Y$ and their associated rotations $R_z, R_y$:

$$Z = \begin{pmatrix} 1 & 0 \\ 0 & -1 \end{pmatrix}, R_z(\theta) = \exp\left(-i\frac{\theta}{2}Z\right), \quad Y = \begin{pmatrix} 0 & -i \\ i & 0 \end{pmatrix}, R_y(\theta) = \exp\left(-i\frac{\theta}{2}Y\right), \quad (1)$$

for rotation angles $\theta \in \mathbb{R}$, and the 2-qubit Ctrl-$Z$ gate $\mathbf{!} = \text{diag}(1, 1, 1, -1)$.

A projective measurement is described by a Hermitian operator $O$ called an observable. Its spectral decomposition $O = \sum_m \alpha_m P_m$ in terms of eigenvalues $\alpha_m$ and orthogonal projections $P_m$ defines the outcomes of this measurement, according to the Born rule: a measured state $|\psi\rangle$ gives the outcome $\alpha_m$ and gets projected onto the state $P_m |\psi\rangle / \sqrt{p(m)}$ with probability $p(m) = \langle\psi| P_m |\psi\rangle = \langle P_m \rangle_\psi$. The expectation value of the observable $O$ with respect to $|\psi\rangle$ is $\mathbb{E}_\psi[O] = \sum_m p(m)\alpha_m = \langle O \rangle_\psi$.

### 2.2 The RAW-PQC and SOFTMAX-PQC policies

At the core of our parametrized quantum policies is a PQC defined by a unitary $U(s, \boldsymbol{\theta})$ that acts on a fixed $n$-qubit state (e.g., $|0^{\otimes n}\rangle$). This unitary encodes an input state $s \in \mathbb{R}^d$ and is parametrized by a trainable vector $\boldsymbol{\theta}$. Although different choices of PQCs are possible, throughout our numerical experiments (Sec. 3 and 4.2), we consider so-called hardware-efficient PQCs [40] with an alternating-layered architecture [28, 41]. This architecture is depicted in Fig. 2 and essentially consists in an alternation of $D_{\text{enc}}$ encoding unitaries $U_{\text{enc}}$ (composed of single-qubit rotations $R_z, R_y$) and $D_{\text{enc}} + 1$ variational unitaries $U_{\text{var}}$ (composed of single-qubit rotations $R_z, R_y$ and entangling Ctrl-$Z$ gates $\mathbf{!}$).

For any given PQC, we define two families of policies, differing in how the final quantum states $|\psi_{s,\boldsymbol{\theta}}\rangle = U(s, \boldsymbol{\theta}) |0^{\otimes n}\rangle$ are used. In the RAW-PQC model, we exploit the probabilistic nature of quantum measurements to define an RL policy. For $|A|$ available actions to the RL agent, we partition $\mathcal{H}$ in $|A|$ disjoint subspaces (e.g., spanned by computational basis states) and associate a projection $P_a$ to each of these subspaces. The projective measurement associated to the observable $O = \sum_a aP_a$ then defines our RAW-PQC policy $\pi_{\boldsymbol{\theta}}(a|s) = \langle P_a \rangle_{s,\boldsymbol{\theta}}$. A limitation of this policy family however is that it does not have a directly adjustable greediness (i.e., a control parameter that makes the policy more peaked). This consideration arises naturally in an RL context where an agent's policy needs to shift from an exploratory behavior (i.e., close to uniform distribution) to a more exploitative

behavior (i.e., a peaked distribution). To remedy this limitation, we define the SOFTMAX-PQC model, that applies an adjustable softmax$_\beta$ non-linear activation function on the expectation values $\langle P_a \rangle_{s,\boldsymbol{\theta}}$ measured on $|\psi_{s,\boldsymbol{\theta}}\rangle$. Since the softmax function normalizes any real-valued input, we can generalize the projections $P_a$ to be arbitrary Hermitian operators $O_a$. We also generalize these observables one step further by assigning them trainable weights. The two models are formally defined below.

**Definition 1** (RAW- and SOFTMAX-PQC). *Given a PQC acting on $n$ qubits, taking as input a state $s \in \mathbb{R}^d$, rotation angles $\boldsymbol{\phi} \in [0, 2\pi]^{|\boldsymbol{\phi}|}$ and scaling parameters $\boldsymbol{\lambda} \in \mathbb{R}^{|\boldsymbol{\lambda}|}$, such that its corresponding unitary $U(s, \boldsymbol{\phi}, \boldsymbol{\lambda})$ produces the quantum state $|\psi_{s,\boldsymbol{\phi},\boldsymbol{\lambda}}\rangle = U(s, \boldsymbol{\phi}, \boldsymbol{\lambda}) |0^{\otimes n}\rangle$, we define its associated RAW-PQC policy as:*

$$\pi_{\boldsymbol{\theta}}(a|s) = \langle P_a \rangle_{s,\boldsymbol{\theta}} \tag{2}$$

*where $\langle P_a \rangle_{s,\boldsymbol{\theta}} = \langle \psi_{s,\boldsymbol{\phi},\boldsymbol{\lambda}} | P_a | \psi_{s,\boldsymbol{\phi},\boldsymbol{\lambda}} \rangle$ is the expectation value of a projection $P_a$ associated to action $a$, such that $\sum_a P_a = I$ and $P_a P_{a'} = \delta_{a,a'}$. $\boldsymbol{\theta} = (\boldsymbol{\phi}, \boldsymbol{\lambda})$ constitute all of its trainable parameters. Using the same PQC, we also define a SOFTMAX-PQC policy as:*

$$\pi_{\boldsymbol{\theta}}(a|s) = \frac{e^{\beta \langle O_a \rangle_{s,\boldsymbol{\theta}}}}{\sum_{a'} e^{\beta \langle O_{a'} \rangle_{s,\boldsymbol{\theta}}}} \tag{3}$$

*where $\langle O_a \rangle_{s,\boldsymbol{\theta}} = \langle \psi_{s,\boldsymbol{\phi},\boldsymbol{\lambda}} | \sum_i w_{a,i} H_{a,i} | \psi_{s,\boldsymbol{\phi},\boldsymbol{\lambda}} \rangle$ is the expectation value of the weighted Hermitian operators $H_{a,i}$ associated to action $a$, $\beta \in \mathbb{R}$ is an inverse-temperature parameter and $\boldsymbol{\theta} = (\boldsymbol{\phi}, \boldsymbol{\lambda}, \boldsymbol{w})$.*

Note that we adopt here a very general definition for the observables $O_a$ of our SOFTMAX-PQC policies. As we discuss in more detail in Appendix C, very expressive trainable observables can in some extreme cases take over all training of the PQC parameters $\boldsymbol{\phi}, \boldsymbol{\lambda}$ and render the role of the PQC in learning trivial. However, in practice, as well as in our numerical experiments, we only consider very restricted observables $O_a = \sum_i w_{a,i} H_{a,i}$, where $H_{a,i}$ are (tensor products of) Pauli matrices or high-rank projections on computational basis states, which do not allow for these extreme scenarios.

In our PQC construction, we include trainable *scaling parameters* $\boldsymbol{\lambda}$, used in every encoding gate to re-scale its input components. This modification to the standard data encoding in PQCs comes in light of recent considerations on the structure of PQC functions [42]. These additional parameters allow to represent functions with a wider and richer spectrum of frequencies, and hence provide shallow PQCs with more expressive power.

## 2.3 Learning algorithm

In order to analyze the properties of our PQC policies without the interference of other learning mechanisms [43], we train these policies using the basic Monte Carlo policy gradient algorithm REINFORCE [44, 45] (see Alg. 1). This algorithm consists in evaluating Monte Carlo estimates of the value function $V_{\pi_{\boldsymbol{\theta}}}(s_0) = \mathbb{E}_{\pi_{\boldsymbol{\theta}}}\left[\sum_{t=0}^{H-1} \gamma^t r_t\right], \gamma \in [0, 1]$, using batches of interactions with the environment, and updating the policy parameters $\boldsymbol{\theta}$ via a gradient ascent on $V_{\pi_{\boldsymbol{\theta}}}(s_0)$. The resulting updates (see line 8 of Alg. 1) involve the gradient of the log-policy $\nabla_{\boldsymbol{\theta}} \log \pi_{\boldsymbol{\theta}}(a|s)$, which we therefore need to compute for our policies. We describe this computation in the following lemma.

**Lemma 1.** *Given a SOFTMAX-PQC policy $\pi_{\boldsymbol{\theta}}$, the gradient of its logarithm is given by:*

$$\nabla_{\boldsymbol{\theta}} \log \pi_{\boldsymbol{\theta}}(a|s) = \beta \left( \nabla_{\boldsymbol{\theta}} \langle O_a \rangle_{s,\boldsymbol{\theta}} - \sum_{a'} \pi_{\boldsymbol{\theta}}(a'|s) \nabla_{\boldsymbol{\theta}} \langle O_{a'} \rangle_{s,\boldsymbol{\theta}} \right). \tag{4}$$

*Partial derivatives with respect to observable weights are trivially given by $\partial_{w_{a,i}} \langle O_a \rangle_{s,\boldsymbol{\theta}} = \langle \psi_{s,\boldsymbol{\phi},\boldsymbol{\lambda}} | H_{a,i} | \psi_{s,\boldsymbol{\phi},\boldsymbol{\lambda}} \rangle$ (see Def. 1), while derivatives with respect to rotation angles $\partial_{\phi_i} \langle O_a \rangle_{s,\boldsymbol{\theta}}$ and scaling parameters[1] $\partial_{\lambda_i} \langle O_a \rangle_{s,\boldsymbol{\theta}}$ can be estimated via the parameter-shift rule [46, 42]:*

$$\partial_i \langle O_a \rangle_{s,\boldsymbol{\theta}} = \frac{1}{2} \left( \langle O_a \rangle_{s,\boldsymbol{\theta} + \frac{\pi}{2} \boldsymbol{e_i}} - \langle O_a \rangle_{s,\boldsymbol{\theta} - \frac{\pi}{2} \boldsymbol{e_i}} \right), \tag{5}$$

*i.e., using the difference of two expectation values $\langle O_a \rangle_{s,\boldsymbol{\theta}'}$ with a single angle shifted by $\pm \frac{\pi}{2}$. For a RAW-PQC policy $\pi_{\boldsymbol{\theta}}$, we have instead:*

$$\nabla_{\boldsymbol{\theta}} \log \pi_{\boldsymbol{\theta}}(a|s) = \nabla_{\boldsymbol{\theta}} \langle P_a \rangle_{s,\boldsymbol{\theta}} / \langle P_a \rangle_{s,\boldsymbol{\theta}} \tag{6}$$

*where the partial derivatives $\partial_{\phi_i} \langle P_a \rangle_{s,\boldsymbol{\theta}}$ and $\partial_{\lambda_i} \langle P_a \rangle_{s,\boldsymbol{\theta}}$ can be estimated similarly to above.*

---

[1] Note that the parameters $\boldsymbol{\lambda}$ do not act as rotation angles. To compute the derivatives $\partial_{\lambda_{i,j}} \langle O_a \rangle_{s,\boldsymbol{\theta}}$, one should compute derivatives w.r.t. $s_j \lambda_{i,j}$ instead and apply the chain rule: $\partial_{\lambda_{i,j}} \langle O_a \rangle_{s,\boldsymbol{\theta}} = s_j \partial_{s_j \lambda_{i,j}} \langle O_a \rangle_{s,\boldsymbol{\theta}}$.

---

**Algorithm 1:** REINFORCE with PQC policies and value-function baselines

---

**Input:** a PQC policy $\pi_{\boldsymbol{\theta}}$ from Def. 1; a value-function approximator $\widetilde{V}_{\boldsymbol{\omega}}$

1   Initialize parameters $\boldsymbol{\theta}$ and $\boldsymbol{\omega}$;

2   **while** *True* **do**

3      Generate $N$ episodes $\{(s_0, a_0, r_1, \ldots, s_{H-1}, a_{H-1}, r_H)\}_i$ following $\pi_{\boldsymbol{\theta}}$;

4      **for** *episode $i$ in batch* **do**

5          Compute the returns $G_{i,t} \leftarrow \sum_{t'=1}^{H-t} \gamma^{t'} r_{t+t'}^{(i)}$;

6          Compute the gradients $\nabla_{\boldsymbol{\theta}} \log \pi_{\boldsymbol{\theta}}(a_t^{(i)}|s_t^{(i)})$ using Lemma 1;

7      Fit $\left\{\widetilde{V}_{\boldsymbol{\omega}}(s_t^{(i)})\right\}_{i,t}$ to the returns $\{G_{i,t}\}_{i,t}$;

8      Compute $\Delta\boldsymbol{\theta} = \frac{1}{N} \sum\limits_{i=1}^{N} \sum\limits_{t=0}^{H-1} \nabla_{\boldsymbol{\theta}} \log \pi_{\boldsymbol{\theta}}(a_t^{(i)}|s_t^{(i)}) \left(G_{i,t} - \widetilde{V}_{\boldsymbol{\omega}}(s_t^{(i)})\right)$;

9      Update $\boldsymbol{\theta} \leftarrow \boldsymbol{\theta} + \alpha\Delta\boldsymbol{\theta}$;

---

In some of our environments, we additionally rely on a linear value-function baseline to reduce the variance of the Monte Carlo estimates [47]. We choose it to be identical to that of Ref. [48].

## 2.4   Efficient policy sampling and policy-gradient evaluation

A natural consideration when it comes to the implementation of our PQC policies is whether one can efficiently (in the number of executions of the PQC on a quantum computer) sample and train them.

By design, sampling from our RAW-PQC policies can be done with a single execution (and measurement) of the PQC: the projective measurement corresponding to the observable $O = \sum_a a P_a$ naturally samples a basis state associated to action $a$ with probability $\langle P_a \rangle_{s,\boldsymbol{\theta}}$. However, as Eq. (6) indicates, in order to train these policies using REINFORCE, one is nonetheless required to estimate the expectation values $\langle P_a \rangle_{s,\boldsymbol{\theta}}$, along with the gradients $\nabla_{\boldsymbol{\theta}} \langle P_a \rangle_{s,\boldsymbol{\theta}}$. Fortunately, these quantities can be estimated efficiently up to some additive error $\varepsilon$, using only $\mathcal{O}(\varepsilon^{-2})$ repeated executions and measurements on a quantum computer.

In the case of our SOFTMAX-PQC policies, it is less clear whether similar noisy estimates $\widetilde{\langle O_a \rangle}_{s,\boldsymbol{\theta}}$ of the expectation values $\langle O_a \rangle_{s,\boldsymbol{\theta}}$ are sufficient to evaluate policies of the form of Eq. (3). We show however that, using these noisy estimates, we can compute a policy $\widetilde{\pi}_{\boldsymbol{\theta}}$ that produces samples close to that of the true policy $\pi_{\boldsymbol{\theta}}$. We state our result formally in the following lemma, proven in Appendix B.

**Lemma 2.** *For a* SOFTMAX-PQC *policy $\pi_{\boldsymbol{\theta}}$ defined by a unitary $U(s, \boldsymbol{\theta})$ and observables $O_a$, call $\widetilde{\langle O_a \rangle}_{s,\boldsymbol{\theta}}$ approximations of the true expectation values $\langle O_a \rangle_{s,\boldsymbol{\theta}}$ with at most $\varepsilon$ additive error. Then the approximate policy $\widetilde{\pi}_{\boldsymbol{\theta}} = \mathrm{softmax}_\beta(\widetilde{\langle O_a \rangle}_{s,\boldsymbol{\theta}})$ has total variation distance $\mathcal{O}(\beta\varepsilon)$ to $\pi_{\boldsymbol{\theta}} = \mathrm{softmax}_\beta(\langle O_a \rangle_{s,\boldsymbol{\theta}})$. Since expectation values can be efficiently estimated to additive error on a quantum computer, this implies efficient approximate sampling from $\pi_{\boldsymbol{\theta}}$.*

We also obtain a similar result for the log-policy gradient of SOFTMAX-PQCs (see Lemma 1), that we show can be efficiently estimated to additive error in $\ell_\infty$-norm (see Appendix B for a proof).

## 3   Performance comparison in benchmarking environments

In the previous section, we have introduced our quantum policies and described several of our design choices. We defined the RAW-PQC and SOFTMAX-PQC models and introduced two original features for PQCs: trainable observables at their output and trainable scaling parameters for their input. In this section, we evaluate the influence of these design choices on learning performance through numerical simulations. We consider three classical benchmarking environments from the OpenAI Gym library [24]: CartPole, MountainCar and Acrobot. All three have continuous state spaces and discrete action spaces (see Appendix D for their specifications). Moreover, simple NN-policies, as well as simple closed-form policies, are known to perform very well in these environments [49], which makes them an excellent test-bed to benchmark PQC policies.

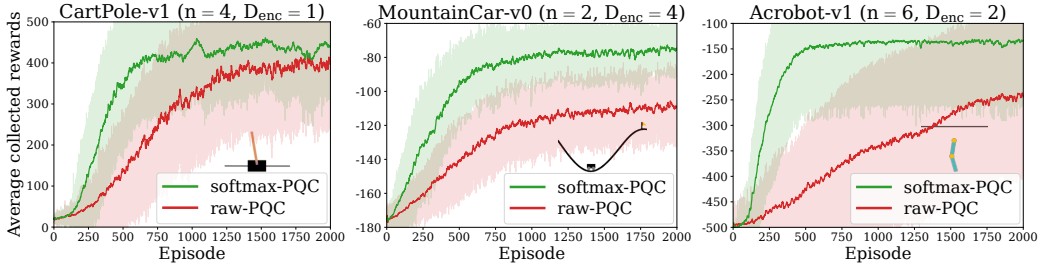

Figure 3: **Numerical evidence of the advantage of SOFTMAX-PQC over RAW-PQC in benchmarking environments.** The learning curves (20 agents per curve) of randomly-initialized SOFTMAX-PQC agents (green curves) and RAW-PQC agents (red curves) in OpenAI Gym environments: CartPole-v1, MountainCar-v0, and Acrobot-v1. Each curve is temporally averaged with a time window of 10 episodes. All agents have been trained using the REINFORCE algorithm (see Alg. 1), with value-function baselines for the MountainCar and Acrobot environments.

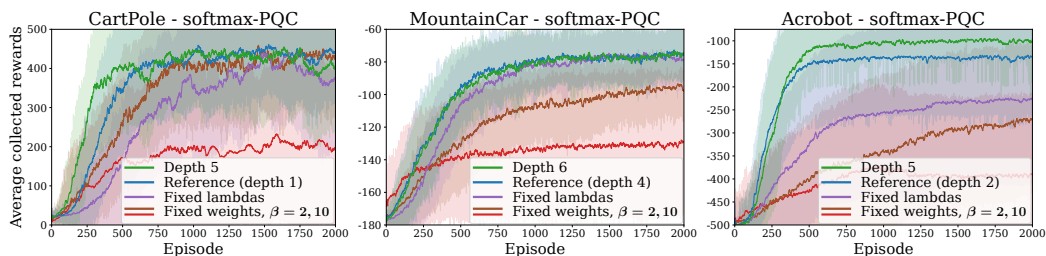

Figure 4: **Influence of the model architecture for SOFTMAX-PQC agents.** The blue curves in each plot correspond to the learning curves from Fig. 3 and are taken as a reference. Other curves highlight the influence of individual hyperparameters. For RAW-PQC agents, see Appendix E.

## 3.1 RAW-PQC v.s. SOFTMAX-PQC

In our first set of experiments, presented in Fig. 3, we evaluate the general performance of our proposed policies. The aim of these experiments is twofold: first, to showcase that quantum policies based on shallow PQCs and acting on very few qubits can be trained to good performance in our selected environments; second, to test the advantage of SOFTMAX-PQC policies over RAW-PQC policies that we conjectured in the Sec. 2.2. To assess these claims, we take a similar approach for each of our benchmarking environments, in which we evaluate the average learning performance of 20 RAW-PQC and 20 SOFTMAX-PQC agents. Apart from the PQC depth, the shared hyperparameters of these two models were jointly picked as to give the best overall performance for both; the hyperparameters specific to each model were optimized independently. As for the PQC depth $D_{enc}$, the latter was chosen as the minimum depth for which near-optimal performance was observed for either model. The simulation results confirm both our hypotheses: quantum policies can achieve good performance on the three benchmarking tasks that we consider, and we can see a clear separation between the performance of SOFTMAX-PQC and RAW-PQC agents.

## 3.2 Influence of architectural choices

The results of the previous subsection however do not indicate whether other design choices we have made in Sec. 2.2 had an influence on the performance of our quantum agents. To address this, we run a second set of experiments, presented in Fig. 4. In these simulations, we evaluate the average performance of our SOFTMAX-PQC agents after modifying one of three design choices: we either increment the depth of the PQC (until no significant increase in performance is observed), fix the input-scaling parameters $\lambda$ to $1$, or fix the observable weights $w$ to $1$. By comparing the performance of these agents with that of the agents from Fig. 3, we can make the following observations:

- **Influence of depth:** Increasing the depth of the PQC generally improves (not strictly) the performance of the agents. Note that the maximum depth we tested was $D_{\text{enc}} = 10$.

- **Influence of scaling parameters $\boldsymbol{\lambda}$:** We observe that training these scaling parameters in general benefits the learning performance of our PQC policies, likely due to their increased expressivity.

- **Influence of trainable observable weights $\boldsymbol{w}$:** our final consideration relates to the importance of having a policy with "trainable greediness" in RL scenarios. For this, we consider SOFTMAX-PQC agents with fixed observables $\beta O_a$ throughout training. We observe that this has the general effect of decreasing the performance and/or the speed of convergence of the agents. We also see that policies with fixed high $\beta$ (or equivalently, a large observable norm $\beta \|O_a\|$) tend to have a poor learning performance, likely due to their lack of exploration in the RL environments.

Finally, note that all the numerical simulations performed here did not include any source of noise in the PQC evaluations. It would be an interesting research direction to assess the influence of (simulated or hardware-induced) noise on the learning performance of PQC agents.

## 4 Quantum advantage of PQC agents in RL environments

The proof-of-concept experiments of the previous section show that our PQC agents can learn in basic classical environments, where they achieve comparable performance to standard DNN policies. This observation naturally raises the question of whether there exist RL environments where PQC policies can provide a learning advantage over standard classical policies. In this section, we answer this question in the affirmative by constructing: a) environments with a provable separation in learning performance between quantum and any classical (polynomial-time) learners, and b) environments where our PQC policies of Sec. 2 show an empirical learning advantage over standard DNN policies.

### 4.1 Quantum advantage of PQC policies over any classical learner

In this subsection, we construct RL environments with theoretical guarantees of separation between quantum and classical learning agents. These constructions are predominantly based on the recent work of Liu *et al.* [14], which defines a classification task out of the discrete logarithm problem (DLP), i.e., the problem solved in the seminal work of Shor [25]. In broad strokes, this task can be viewed as an encryption of an easy-to-learn problem. For an "un-encrypted" version, one defines a labeling $f_s$ of integers between $0$ and $p - 2$ (for a large prime $p$), where the integers are labeled positively if and only if they lie in the segment $[s, s + (p - 3)/2]$ (mod $p - 1$). Since this labeling is linearly separable, the concept class $\{f_s\}_s$ is then easy to learn. To make it hard, the input integers $x$ (now between $1$ and $p - 1$) are first encrypted using modular exponentiation, i.e., the secure operation performed in the Diffie–Hellman key exchange protocol. In the encrypted problem, the logarithm of the input integer $\log_g(x)$ (for a generator $g$ of $\mathbb{Z}_p^*$, see Appendix F) hence determines the label of $x$. Without the ability to decrypt by solving DLP, which is widely believed to be classically intractable, the numbers appear randomly labeled. Moreover, Liu *et al.* show that achieving non-trivial labeling accuracy $1/2 + 1/\text{poly}(n)$ (for $n = \log(p)$, i.e., slightly better than random guessing) with a classical polynomial-time algorithm using $\text{poly}(n)$ examples would lead to an efficient classical algorithm that solves DLP [14]. In contrast, the same authors construct a family of quantum learners based on Shor's algorithm, that can achieve a labeling accuracy larger than $0.99$ with high probability.

**SL-DLP** Our objective is to show that analogous separations between classical and quantum learners can be established for RL environments, in terms of their attainable value functions. We start by pointing out that supervised learning (SL) tasks (and so the classification problem of Liu *et al.*) can be trivially embedded into RL environments [50]: for a given concept $f_s$, the states $x$ are datapoints, an action $a$ is an agent's guess on the label of $x$, an immediate reward specifies if it was correct (i.e., $f_s(x) = a$), and subsequent states are chosen uniformly at random. In such settings, the value function is trivially related to the testing accuracy of the SL problem, yielding a direct reduction of the separation result of Liu *et al.* [14] to an RL setting. We call this family of environments SL-DLP.

**Cliffwalk-DLP** In the SL-DLP construction, we made the environment fully random in order to simulate the process of obtaining i.i.d. samples in an SL setting. It is an interesting question whether similar results can be obtained for environments that are less random, and endowed with temporal structure, which is characteristic of RL. In our second family of environments (Cliffwalk-DLP),

we supplement the SL-DLP construction with next-state transitions inspired by the textbook "cliff walking" environment of Sutton & Barto [44]: all states are ordered in a chain and some actions of the agent can lead to immediate episode termination. We keep however stochasticity in the environment by allowing next states to be uniformly sampled, with a certain probability $\delta$ (common in RL to ensure that an agent is not simply memorizing a correct sequence of actions). This allows us to show that, as long as sufficient randomness is provided, we still have a simple classical-quantum separation.

**Deterministic-DLP**  In the two families constructed above, each environment instance provided the randomness needed for a reduction from the SL problem. This brings us to the question of whether separations are also possible for fully deterministic environments. In this case, it is clear that for any given environment, there exists an efficient classical agent which performs perfectly over any polynomial horizon (a lookup-table will do). However, we show in our third family of environments (Deterministic-DLP) that a separation can still be attained by moving the randomness to the choice of the environment itself: assuming an efficient classical agent is successful in most of exponentially-many randomly generated (but otherwise deterministic) environments, implies the existence of a classical efficient algorithm for DLP.

We summarize our results in the following theorem, detailed and proven in Appendices G through I.

**Theorem 1.** *There exist families of reinforcement learning environments which are: i) fully random (i.e., subsequent states are independent from the previous state and action); ii) partially random (i.e., the previous moves determine subsequent states, except with a probability $\delta$ at least $0.86$ where they are chosen uniformly at random), and iii) fully deterministic; such that there exists a separation in the value functions achievable by a given quantum polynomial-time agent and any classical polynomial-time agent. Specifically, the value of the initial state for the quantum agent $V_q(s_0)$ is $\varepsilon-$close to the optimal value function (for a chosen $\varepsilon$, and with probability above 2/3). Further, if there exists a classical efficient learning agent that achieves a value $V_c(s_0)$ better than $V_{rand}(s_0) + \varepsilon'$ (for a chosen $\varepsilon'$, and with probability above 0.845), then there exists a classical efficient algorithm to solve DLP. Finally, we have $V_q(s_0) - V_c(s_0)$ larger than some constant, which depends on the details of the environment.*

The remaining point we need to address here is that the learning agents of Liu *et al.* do not rely on PQCs but rather support vector machines (SVMs) based on quantum kernels [6, 7]. Nonetheless, using a connection between these quantum SVMs and PQCs [7], we construct PQC policies which are as powerful in solving the DLP environments as the agents of Liu *et al.* (even under similar noise considerations). We state our result in the following informal theorem, that we re-state formally, along with the details of our construction in Appendices J and K.

**Theorem 2** (informal version)**.** *Using a training set of size polynomial in $n = \log(p)$ and a number of (noisy) quantum circuit evaluations also polynomial in $n$, we can train a PQC classifier on the DLP task of Liu* et al. *of size $n$ that achieves a testing accuracy arbitrarily close to optimal, with high probability. This PQC classifier can in turn be used to construct close-to-optimal quantum agents in our DLP environments, as prescribed by Theorem 1.*

### 4.2   Quantum advantage of PQC policies over DNN policies

While the DLP environments establish a proof of the learning advantage PQC policies can have in theory, these environments remain extremely contrived and artificial. They are based on algebraic properties that agents must explicitly decrypt in order to perform well. Instead, we would like to consider environments that are less tailored to a specific decryption function, which would allow more general agents to learn. To do this, we take inspiration from the work of Havlíček *et al.* [6], who, in order to test their PQC classifiers, define a learning task generated by similar quantum circuits.

#### 4.2.1   PQC-generated environments

We generate our RL environments out of random RAW-PQCs. To do so, we start by uniformly sampling a RAW-PQC that uses the alternating-layer architecture of Fig. 2 for $n = 2$ qubits and depth $D_{enc} = 4$. We use this RAW-PQC to generate a labeling function $f(s)$ by assigning a label $+1$ to the datapoints $s$ in $[0, 2\pi]^2$ for which $\langle ZZ \rangle_{s,\boldsymbol{\theta}} \geq 0$ and a label $-1$ otherwise. We create a dataset $S$ of 10 datapoints per label by uniformly sampling points in $[0, 2\pi]^2$ for which $|\langle ZZ \rangle_{s,\boldsymbol{\theta}}| \geq \frac{\Delta}{2} = 0.15$.

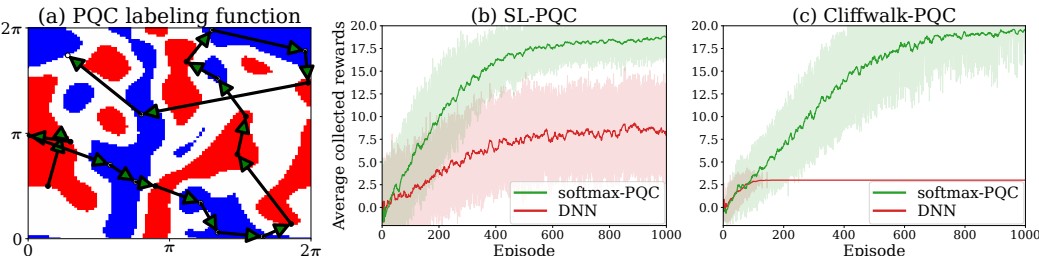

Figure 5: **Numerical evidence of the advantage of PQC policies over DNN policies in PQC-generated environments.** (a) Labeling function and training data used for both RL environments. The data labels (red for $+1$ label and blue for $-1$ label) are generated using a RAW-PQC of depth $D_{\text{enc}} = 4$ with a margin $\Delta = 0.3$ (white areas). The training samples are uniformly sampled from the blue and red regions, and arrows indicate the rewarded path of the cliffwalk environment. (b) and (c) The learning curves (20 agents per curve) of randomly-initialized SOFTMAX-PQC agents and DNN agents in RL environments where input states are (b) uniformly sampled from the dataset and (c) follow cliffwalk dynamics. Each curve is temporally averaged with a time window of 10 episodes.

This dataset allows us to define two RL environments, similar to the SL-DLP and Cliffwalk-DLP environments of Sec. 4.1:

- **SL-PQC:** this degenerate RL environment encodes a classification task in an episodic RL environment: at each interaction step of a 20-step episode, a sample state $s$ is uniformly sampled from the dataset $S$, the agent assigns a label $a = \pm 1$ to it and receives a reward $\delta_{f(s),a} = \pm 1$.
- **Cliffwalk-PQC:** this environment essentially adds a temporal structure to SL-PQC: each episode starts from a fixed state $s_0 \in S$, and if an agent assigns the correct label to a state $s_i$, $0 \leq i \leq 19$, it moves to a fixed state $s_{i+1}$ and receives a $+1$ reward, otherwise the episode is instantly terminated and the agent gets a $-1$ reward. Reaching $s_{20}$ also causes termination.

### 4.2.2 Performance comparison

Having defined our PQC-generated environments, we now evaluate the performance of SOFTMAX-PQC and DNN policies in these tasks. The particular models we consider are SOFTMAX-PQCs with PQCs sampled from the same family as that of the RAW-PQCs generating the environments (but with re-initialized parameters $\boldsymbol{\theta}$), and DNNs using Rectified Linear Units (ReLUs) in their hidden layers. In our hyperparameter search, we evaluated the performance of DNNs with a wide range of depths (number of hidden layers between 2 to 10) and widths (number of units per hidden layer between 8 and 64), and kept the architecture with the best average performance (depth 4, width 16).

Despite this hyperparametrization, we find (see Fig. 5, and Fig. 8 in Appendix E for different environment instances) that the performance of DNN policies on these tasks remains limited compared to that of SOFTMAX-PQCs, that learn close-to-optimal policies on both tasks. Moreover, we observe that the separation in performance gets boosted by the cliffwalk temporal structure. This is likely do to the increased complexity of this task, as, in order to move farther in the cliffwalk, the policy family should allow learning new labels without "forgetting" the labels of earlier states. In these particular case studies, the SOFTMAX-PQC policies exhibited sufficient flexibility in this sense, whereas the DNNs we considered did not (see Appendix E for a visualization of these policies). Note that these results do not reflect the difficulty of our tasks at the sizes we consider (a look-up table would perform optimally) but rather highlight the inefficacy of these DNNs at learning PQC functions.

# 5 Conclusion

In this work, we have investigated the design of quantum RL agents based on PQCs. We proposed several constructions and showed the impact of certain design choices on learning performance. In particular, we introduced the SOFTMAX-PQC model, where a softmax policy is computed from expectation values of a PQC with both trainable observable weights and input scaling parameters. These added features to standard PQCs used in ML (e.g., as quantum classifiers) enhance both the expressivity and flexibility of PQC policies, which allows them to achieve a learning performance on benchmarking environments comparable to that of standard DNNs. We additionally demonstrated the existence of task environments, constructed out of PQCs, that are very natural for PQC agents, but on which DNN agents have a poor performance. To strengthen this result, we constructed several RL environments, each with a different degree of degeneracy (i.e., closeness to a supervised learning task), where we showed a rigorous separation between a class of PQC agents and any classical learner, based on the widely-believed classical hardness of the discrete logarithm problem. We believe that our results constitute strides toward a practical quantum advantage in RL using near-term quantum devices.

# 6 Broad impact

We expect our work to have an overall positive societal impact. Notably, we believe that our approach to QRL could be beneficial in the two following ways:

- Modern-day RL is known to be very resource-heavy in terms of compute power and energy consumption (see, e.g., the resources needed to train AlphaGo Zero [17]). In other computational problems, e.g., the quantum supremacy problem of Google [51], it was shown that, because of their computational advantages, quantum computers could save many orders of magnitude in energy consumption compared to classical supercomputers [52]. Therefore, a quantum learning advantage as showcased in our work could potentially alleviate the computational demands of RL, making it more economically appealing and environmentally-friendly.

- Aside from the game-based problems that we consider in our work, the areas of application of RL are constantly increasing [53–55]. The learning advantages of QRL could potentially make these existing applications more accessible technologically and economically, but also unlock new applications, e.g., in problems in quantum information [56, 22] or quantum chemistry [57].

At the same time, our work may have certain negative consequences. Notably, QRL will inherit many of the problems that are already present in classical RL and ML in general. For instance, it is not clear whether the question of interpretability of learning models [58] will be negatively or positively impacted by switching to quantum models. One could argue that the inability to fully access the quantum Hilbert spaces in which quantum computers operate can turn learning models even further into "black-boxes" than existing classical models. Also, similarly to the fact that current state-of-the-art ML/RL requires supercomputers that are not accessible to everyone, private and select access to quantum computers could emphasize existing inequalities in developing and using AI.

## Acknowledgments and Disclosure of Funding

The authors would like to thank Srinivasan Arunachalam for clarifications on the testing accuracy of their quantum classifier in the DLP classification task. The authors would also like to thank Andrea Skolik and Arjan Cornelissen for helpful discussions and comments. CG thanks Thomas Moerland for discussions in the early phases of this project. SJ and HJB acknowledge support from the Austrian Science Fund (FWF) through the projects DK-ALM:W1259-N27 and SFB BeyondC F7102. SJ also acknowledges the Austrian Academy of Sciences as a recipient of the DOC Fellowship. This work was in part supported by the Dutch Research Council (NWO/OCW), as part of the Quantum Software Consortium program (project number 024.003.037). VD and SM acknowledge the support by the project NEASQC funded from the European Union's Horizon 2020 research and innovation programme (grant agreement No 951821). VD and SM also acknowledge partial funding by an unrestricted gift from Google Quantum AI. The computational results presented here have been achieved in part using the LEO HPC infrastructure of the University of Innsbruck.

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
