# Supplementary Material for:
## Parametrized Quantum Policies for Reinforcement Learning

**Sofiene Jerbi, Casper Gyurik, Simon C. Marshall, Hans J. Briegel, Vedran Dunjko**

**Outline** The Supplementary Material is organized as follows. In Appendix A, we derive the expression of the log-policy gradient for SOFTMAX-PQCs presented in Lemma 1. In Appendix B, we prove Lemmas 2 and 3 on the efficient policy sampling and the efficient estimation of the log-policy gradient for SOFTMAX-PQC policies. In Appendix C, we clarify the role of the trainable observables in our definition of SOFTMAX-PQC policies. In Appendix D, we give a specification of the environments considered in our numerical simulations, as well the hyperparameters we used to train all RL agents. In Appendix E, we present additional plots and numerical simulations that help our understanding and visualization of PQC polices. In Appendix F, we give a succinct description of the DLP classification task of Liu *et al.* In Appendices G to I, we prove our main Theorem 1 on learning separations in DLP environments. In appendix J, we construct PQC agents with provable guarantees of solving the DLP environments, stated and proven in Theorem 2.

## A    Derivation of the log-policy gradient

For a SOFTMAX-PQC defined in Def. 1, we have:

$$
\nabla_{\boldsymbol{\theta}} \log \pi_{\boldsymbol{\theta}}(a|s) = \nabla_{\boldsymbol{\theta}} \log e^{\beta \langle O_a \rangle_{s,\boldsymbol{\theta}}} - \nabla_{\boldsymbol{\theta}} \log \sum_{a'} e^{\beta \langle O_{a'} \rangle_{s,\boldsymbol{\theta}}}
$$

$$
= \beta \nabla_{\boldsymbol{\theta}} \langle O_a \rangle_{s,\boldsymbol{\theta}} - \sum_{a'} \frac{e^{\beta \langle O_{a'} \rangle_{s,\boldsymbol{\theta}}} \beta \nabla_{\boldsymbol{\theta}} \langle O_{a'} \rangle_{s,\boldsymbol{\theta}}}{\sum_{a''} e^{\beta \langle O_{a''} \rangle_{s,\boldsymbol{\theta}}}}
$$

$$
= \beta \left( \nabla_{\boldsymbol{\theta}} \langle O_a \rangle_{s,\boldsymbol{\theta}} - \sum_{a'} \pi_{\boldsymbol{\theta}}(a'|s) \nabla_{\boldsymbol{\theta}} \langle O_{a'} \rangle_{s,\boldsymbol{\theta}} \right).
$$

## B    Efficient implementation of SOFTMAX-PQC policies

### B.1    Efficient approximate policy sampling

In this section we prove Lemma 2, restated below:

**Lemma 2.** *For a* SOFTMAX-PQC *policy $\pi_{\boldsymbol{\theta}}$ defined by a unitary $U(s, \boldsymbol{\theta})$ and observables $O_a$, call $\langle \widetilde{O_a} \rangle_{s,\boldsymbol{\theta}}$ approximations of the true expectation values $\langle O_a \rangle_{s,\boldsymbol{\theta}}$ with at most $\varepsilon$ additive error. Then the approximate policy $\widetilde{\pi}_{\boldsymbol{\theta}} = \mathrm{softmax}_{\beta}(\langle \widetilde{O_a} \rangle_{s,\boldsymbol{\theta}})$ has total variation distance $\mathcal{O}(\beta \varepsilon)$ to $\pi_{\boldsymbol{\theta}} = \mathrm{softmax}_{\beta}(\langle O_a \rangle_{s,\boldsymbol{\theta}})$. Since expectation values can be efficiently estimated to additive error on a quantum computer, this implies efficient approximate sampling from $\pi_{\boldsymbol{\theta}}$.*

*Proof.* Consider $|A|$ estimates $\left\{ \langle \widetilde{O_a} \rangle_{s,\boldsymbol{\theta}} \right\}_{1 \le a \le |A|}$, obtained all to additive error $\varepsilon$, i.e.,

$$
\left| \langle \widetilde{O_a} \rangle_{s,\boldsymbol{\theta}} - \langle O_a \rangle_{s,\boldsymbol{\theta}} \right| \le \varepsilon, \quad \forall a
$$

and used to compute an approximate policy

$$
\widetilde{\pi}_{\boldsymbol{\theta}}(a|s) = \frac{e^{\beta \langle \widetilde{O_a} \rangle_{s,\boldsymbol{\theta}}}}{\sum_{a'} e^{\beta \langle \widetilde{O_{a'}} \rangle_{s,\boldsymbol{\theta}}}}.
$$

Due to the monoticity of the exponential, we have, for all $a$:

$$\frac{e^{-\beta\varepsilon}e^{\beta\langle O_a\rangle_{s,\boldsymbol{\theta}}}}{e^{\beta\varepsilon}\sum_{a'}e^{\beta\langle O_{a'}\rangle_{s,\boldsymbol{\theta}}}} \leq \frac{e^{\beta\langle\widetilde{O_a}\rangle_{s,\boldsymbol{\theta}}}}{\sum_{a'}e^{\beta\langle\widetilde{O_{a'}}\rangle_{s,\boldsymbol{\theta}}}} \leq \frac{e^{\beta\varepsilon}e^{\beta\langle O_a\rangle_{s,\boldsymbol{\theta}}}}{e^{-\beta\varepsilon}\sum_{a'}e^{\beta\langle O_{a'}\rangle_{s,\boldsymbol{\theta}}}}$$

$$\Leftrightarrow \quad e^{-2\beta\varepsilon}\pi_{\boldsymbol{\theta}}(a|s) \leq \quad \widetilde{\pi}_{\boldsymbol{\theta}}(a|s) \quad \leq e^{2\beta\varepsilon}\pi_{\boldsymbol{\theta}}(a|s). \tag{7}$$

Hence,

$$\begin{aligned}
\mathrm{TV}(\pi_{\boldsymbol{\theta}}, \widetilde{\pi}_{\boldsymbol{\theta}}) &= \sum_a |\widetilde{\pi}_{\boldsymbol{\theta}}(a|s) - \pi_{\boldsymbol{\theta}}(a|s)| \\
&\leq \sum_a \left| e^{2\beta\varepsilon}\pi_{\boldsymbol{\theta}}(a|s) - e^{-2\beta\varepsilon}\pi_{\boldsymbol{\theta}}(a|s) \right| \\
&= \sum_a \left| e^{2\beta\varepsilon} - e^{-2\beta\varepsilon} \right| \pi_{\boldsymbol{\theta}}(a|s) \\
&= 2|\sinh(2\beta\varepsilon)| \underset{\beta\varepsilon\to 0^+}{=} 4\beta\varepsilon + \mathcal{O}\left((\beta\varepsilon)^3\right),
\end{aligned}$$

where we used $\{\widetilde{\pi}_{\boldsymbol{\theta}}(a|s), \pi_{\boldsymbol{\theta}}(a|s)\} \in [e^{-2\beta\varepsilon}\pi_{\boldsymbol{\theta}}(a|s), e^{2\beta\varepsilon}\pi_{\boldsymbol{\theta}}(a|s)]$ in the first inequality. $\qquad\square$

### B.2 Efficient estimation of the log-policy gradient

Using a similar approach to the proof of the previous section, we show the following lemma:

**Lemma 3.** *For a* SOFTMAX-PQC *policy $\pi_{\boldsymbol{\theta}}$ defined by a unitary $U(s,\boldsymbol{\theta})$ and observables $O_a$, call $\partial_i\langle\widetilde{O_a}\rangle_{s,\boldsymbol{\theta}}$ approximations of the true derivatives $\partial_i\langle O_a\rangle_{s,\boldsymbol{\theta}}$ with at most $\varepsilon$ additive error, and $\langle\widetilde{O_a}\rangle_{s,\boldsymbol{\theta}}$ approximations of the true expectation values $\langle O_a\rangle_{s,\boldsymbol{\theta}}$ with at most $\varepsilon' = \varepsilon(4\beta\max_a\|O_a\|)^{-1}$ additive error. Then the approximate log-policy gradient $\nabla_{\boldsymbol{\theta}}\log\widetilde{\pi}_{\boldsymbol{\theta}}(a|s) = \beta\left(\nabla_{\boldsymbol{\theta}}\langle\widetilde{O_a}\rangle_{s,\boldsymbol{\theta}} - \sum_{a'}\widetilde{\pi}_{\boldsymbol{\theta}}(a'|s)\nabla_{\boldsymbol{\theta}}\langle\widetilde{O_{a'}}\rangle_{s,\boldsymbol{\theta}}\right)$ has distance $\mathcal{O}(\beta\varepsilon)$ to $\nabla_{\boldsymbol{\theta}}\log\pi_{\boldsymbol{\theta}}(a|s)$ in $\ell_\infty$-norm.*

*Proof.* Call $x_{a,i} = \pi_{\boldsymbol{\theta}}(a|s)\partial_i\langle O_a\rangle_{s,\boldsymbol{\theta}}$ and $\widetilde{x}_{a,i} = \widetilde{\pi}_{\boldsymbol{\theta}}(a|s)\partial_i\langle\widetilde{O_a}\rangle_{s,\boldsymbol{\theta}}$, such that:

$$\partial_i\log\widetilde{\pi}_{\boldsymbol{\theta}}(a|s) = \beta\left(\partial_i\langle\widetilde{O_a}\rangle_{s,\boldsymbol{\theta}} - \sum_{a'}\widetilde{x}_{a',i}\right).$$

and similarly for $\partial_i\log\pi_{\boldsymbol{\theta}}(a|s)$.

Using Eq. (7) and that $|\partial_i\langle O_a\rangle_{s,\boldsymbol{\theta}} - \partial_i\langle\widetilde{O_a}\rangle_{s,\boldsymbol{\theta}}| \leq \varepsilon, \forall a, i$, we have:

$$e^{-2\beta\varepsilon'}\pi_{\boldsymbol{\theta}}(a|s)\left(\partial_i\langle O_a\rangle_{s,\boldsymbol{\theta}} - \varepsilon\right) \leq \widetilde{\pi}_{\boldsymbol{\theta}}(a|s)\partial_i\langle\widetilde{O_a}\rangle_{s,\boldsymbol{\theta}} \leq e^{2\beta\varepsilon'}\pi_{\boldsymbol{\theta}}(a|s)\left(\partial_i\langle O_a\rangle_{s,\boldsymbol{\theta}} + \varepsilon\right)$$

$$\Rightarrow \quad e^{-2\beta\varepsilon'}\left(\sum_a x_{a,i} - \varepsilon\right) \leq \quad \sum_a \widetilde{x}_{a,i} \quad \leq e^{2\beta\varepsilon'}\left(\sum_a x_{a,i} + \varepsilon\right)$$

where we summed the first inequalities over all $a$. Hence:

$$\begin{aligned}
\left|\sum_a x_{a,i} - \sum_a \widetilde{x}_{a,i}\right| &\leq \left| e^{2\beta\varepsilon'}\left(\sum_a x_{a,i} + \varepsilon\right) - e^{-2\beta\varepsilon'}\left(\sum_a x_{a,i} - \varepsilon\right) \right| \\
&\leq \left| (e^{2\beta\varepsilon'} + e^{-2\beta\varepsilon'})\varepsilon + (e^{2\beta\varepsilon'} - e^{-2\beta\varepsilon'})\sum_a x_{a,i} \right| \\
&\leq \left| 2\cosh(2\beta\varepsilon')\varepsilon + 2\sinh(2\beta\varepsilon')\sum_a x_{a,i} \right| \\
&\underset{\beta\varepsilon'\to 0^+}{=} \left| \varepsilon + 4\beta\varepsilon'\sum_a x_{a,i} + \mathcal{O}\left((\beta\varepsilon')^2\varepsilon\right) + \mathcal{O}\left((\beta\varepsilon')^3\right) \right|. \tag{8}
\end{aligned}$$

We also have

$$\left|\sum_a x_{a,i}\right| = \left|\sum_a \pi_{\boldsymbol{\theta}}(a|s)\partial_i\langle O_a\rangle_{s,\boldsymbol{\theta}}\right| \leq \max_{a,i}|\partial_i\langle O_a\rangle_{s,\boldsymbol{\theta}}| \leq \max_a\|O_a\|$$

where the last inequality derives from the parameter-shift rule (Eq. (5)) formulation of $\partial_i \langle O_a \rangle$ for derivatives w.r.t. rotation angles of the PQC and the fact that $\partial_i \langle O_a \rangle$ are simply expectation values $\langle H_{a,i} \rangle$ with $\|H_{a,i}\| \leq \|O_a\|$ for observable weights.

Applying the triangular inequality on the right side of Eq. (8), we hence have:

$$\left| \sum_a x_{a,i} - \sum_a \widetilde{x}_{a,i} \right| \underset{\beta\varepsilon'\to 0^+}{\leq} \varepsilon + 4\beta\varepsilon' \max_a \|O_a\| + \mathcal{O}\left((\beta\varepsilon')^2\varepsilon\right) + \mathcal{O}\left((\beta\varepsilon')^3\right).$$

For $\varepsilon' = \varepsilon(4\beta \max_a \|O_a\|)^{-1}$ and using $|\partial_i\langle O_a\rangle_{s,\boldsymbol{\theta}} - \partial_i\langle\widetilde{O_a}\rangle_{s,\boldsymbol{\theta}}| \leq \varepsilon, \forall a, i$, we finally have:

$$|\partial_i \log \pi_{\boldsymbol{\theta}}(a|s) - \partial_i \log \widetilde{\pi_{\boldsymbol{\theta}}}(a|s)| \underset{\beta\varepsilon\to 0^+}{\leq} 3\beta\varepsilon + \mathcal{O}(\beta\varepsilon^3) \quad \forall i \qquad \qquad \square$$

## C  The role of trainable observables in SOFTMAX-PQC policies

In Sec. 2.2, we presented a general definition of the SOFTMAX-PQC observables $O_a = \sum_i w_{a,i} H_{a,i}$ in terms of an arbitrary weighted sum of Hermitian matrices $H_{a,i}$. In this appendix, we clarify the role of such a decomposition.

### C.1  Training the eigenbasis and the eigenvalues of an observable

Consider a projective measurement defined by an observable $O = \sum_m \alpha_m P_m$, to be performed on a quantum state of the form $V(\boldsymbol{\theta}) |\psi\rangle$, where $V(\boldsymbol{\theta})$ denotes a (variational) unitary. Equivalently, one could also measure the observable $V^\dagger(\boldsymbol{\theta})OV(\boldsymbol{\theta})$ on the state $|\psi\rangle$. Indeed, these two measurements have the same probabilities $p(m) = \langle\psi| V^\dagger(\boldsymbol{\theta})P_mV(\boldsymbol{\theta}) |\psi\rangle$ of measuring any outcome $\alpha_m$. Note also that the possible outcomes $\alpha_m$ (i.e., the eigenvalues of the observable $O$) remain unchanged.

From this observation, it is then clear that, by defining an observable $O = \sum_m \alpha_m P_m$ using projections $P_m$ on each computational basis state of the Hilbert space $\mathcal{H}$ and arbitrary eigenvalues $\alpha_m \in \mathbb{R}$, the addition of a *universal* variational unitary $V(\boldsymbol{\theta})$ prior to the measurement results in a family of observables $\{V^\dagger(\boldsymbol{\theta})OV(\boldsymbol{\theta})\}_{\boldsymbol{\theta},\boldsymbol{\alpha}}$ that covers all possible Hermitian observables in $\mathcal{H}$. Moreover, in this setting, the parameters that define the eigenbasis of the observables $V^\dagger(\boldsymbol{\theta})OV(\boldsymbol{\theta})$ (i.e., $\boldsymbol{\theta}$) are completely distinct from the parameters that define their eigenvalues (i.e., $\boldsymbol{\alpha}$). This is not the case for observables that are expressed as linear combinations of non-commuting matrices, for instance.

In our simulations, we consider restricted families of observables. In particular, we take the Hermitian matrices $H_{a,i}$ to be diagonal in the computational basis (e.g., tensor products of Pauli-$Z$ matrices), which means they, as well as $O_a$, can be decomposed in terms of projections on the computational basis states. However, the resulting eigenvalues $\boldsymbol{\alpha}$ that we obtain from this decomposition are in our case degenerate, which means that the weights $\boldsymbol{w}_a$ underparametrize the spectrums of the observables $O_a$. Additionally, the last variational unitaries $V_{\mathrm{var}}(\boldsymbol{\phi}_L)$ of our PQCs are far from universal, which restricts the accessible eigenbasis of all variational observables $V_{\mathrm{var}}^\dagger(\boldsymbol{\phi}_L)O_aV_{\mathrm{var}}(\boldsymbol{\phi}_L)$.

### C.2  The power of universal observables

Equivalently to the universal family of observables $\{V^\dagger(\boldsymbol{\theta})OV(\boldsymbol{\theta})\}_{\boldsymbol{\theta},\boldsymbol{\alpha}}$ that we defined in the previous section, one can construct a family of observables $\{O_{\boldsymbol{w}} = \sum_i w_i H_i\}_{\boldsymbol{w}}$ that parametrizes all Hermitian matrices in $\mathcal{H}$ (e.g., by taking $H_i$ to be single components of a Hermitian matrix acting on $\mathcal{H}$). Note that this family is covered by our definition of SOFTMAX-PQC observables. Now, given access to data-dependent quantum states $|\psi_s\rangle$ that are expressive enough (e.g., a binary encoding of the input $s$, or so-called universal quantum feature states [1]), one can approximate arbitrary functions of $s$ using expectations values of the form $\langle\psi_s| O_{\boldsymbol{w}} |\psi_s\rangle$. This is because the observables $O_{\boldsymbol{w}}$ can encode an arbitrary quantum computation. Hence, in the case of our SOFTMAX-PQCs, one could use such observables and such encodings $|\psi_s\rangle$ of the input states $s$ to approximate any policy $\pi(a|s)$ (using an additional softmax), without the need for any variational gates in the PQC generating $|\psi_s\rangle$.

As we mentioned in the previous section, the observables that we consider in this work are more restricted, and moreover, the way we encode the input states $s$ leads to non-trivial encodings $|\psi_{s,\boldsymbol{\phi},\boldsymbol{\lambda}}\rangle$ in general. This implies that the variational parameters $\boldsymbol{\phi}, \boldsymbol{\lambda}$ of our PQCs have in general a non-trivial role in learning good policies. One can even show here that these degrees of freedom are sufficient to make such PQCs universal function approximators [2].

## D   Environments specifications and hyperpameters

In Table 1, we present a specification of the environments we consider in our numerical simulations. These are standard benchmarking environments from the OpenAI Gym library [3], described in Ref. [4], PQC-generated environments that we define in Sec. 4.2, and the CognitiveRadio environment of Ref. [5] that we discuss in Appendix E.

| Environment | State dimension | Number of actions | Horizon | Reward function | Termination conditions |
|---|---|---|---|---|---|
| CartPole-v1 | 4 | 2 | 500 | +1 until termination | • Pole angle or cart position outside of bounds • Reaching horizon |
| MountainCar-v0 | 2 | 3 | 200 | −1 + height until termination | Reaching goal or horizon |
| Acrobot-v1 | 6 | 3 | 500 | −1 until termination | Reaching goal or horizon |
| SL-PQC | 2 | 2 | 20 | +1 for good action −1 for wrong action | Reaching horizon |
| Cliffwalk-PQC | 2 | 2 | 20 | +1 for good action −1 for wrong action | • Doing wrong action • Reaching horizon |
| CognitiveRadio | 2 to 5 (discrete) | 2 to 5 | 100 | +1 for good action −1 for wrong action | Reaching horizon |

Table 1: **Environments specifications.** The reward function of Mountaincar-v0 has been modified compared to the standard specification of OpenAI Gym [3], similarly to Ref. [6].

In Tables 2 and 3, we list the hyperparameters used to train our agents on the various environments we consider. All agents use an ADAM optimizer. For the plots presented in this manuscript, all quantum circuits were implemented using the Cirq library [7] in Python and simulated using a Qulacs backend [8] in C++. For the tutorial [9], the TensorFlow Quantum library [10] was used.
All simulations were run on the LEO cluster (more than 3000 CPUs) of the University of Innsbruck, with an estimated total compute time (including hyperparametrization) of 20 000 CPU-hours.

## E   Deferred plots and shape of policies learned by PQCs v.s. DNNs

### E.1   Influence of architectural choices on RAW-PQC agents

In Fig. 6, we run a similar experiment to that of Sec. 3.2 in the main text, but on RAW-PQC agents instead of SOFTMAX-PQC agents. We observe that both increasing the depth of the PQCs and training the scaling parameters $\lambda$ have a similar positive influence on the learning performance, and even more pronounced than for SOFTMAX-PQC agents. Nonetheless, we also observe that, even at greater depth, the final performance, as well as the speed of convergence, of RAW-PQC agents remain limited compared to that of SOFTMAX-PQC agents.

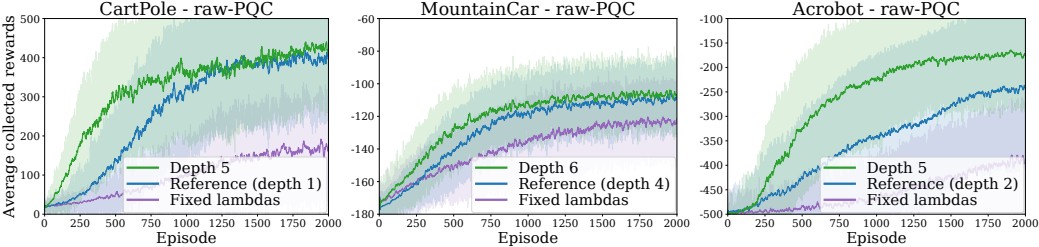

Figure 6: **Influence of the model architecture for RAW-PQC agents.** The blue curves in each plot correspond to the learning curves from Fig. 3 and are taken as a reference.

| Environment | Model | Learning rates | Discount $\gamma$ | Final $\beta$ | Batch size | Depth | Width |
|---|---|---|---|---|---|---|---|
| CartPole-v1 | SOFTMAX-PQC | $[0.01, 0.1, 0.1]$ | 1 | 1 | 10 | $\{1, 5\}$ | 4 |
| | RAW-PQC | $[0.01, 0., 0.1]$ | 1 | ✗ | 10 | $\{1, 5\}$ | 4 |
| MountainCar-v0 | SOFTMAX-PQC | $[0.01, 0.1, 0.01]$ | 1 | 1.5 | 10 | $\{4, 6\}$ | 2 |
| | RAW-PQC | $[0.01, 0., 0.01]$ | 1 | ✗ | 10 | $\{4, 6\}$ | 2 |
| Acrobot-v1 | SOFTMAX-PQC | $[0.01, 0.1, 0.1]$ | 1 | 1 | 10 | $\{2, 5\}$ | 6 |
| | RAW-PQC | $[0.01, 0., 0.1]$ | 1 | ✗ | 10 | $\{2, 5\}$ | 6 |
| SL-PQC | SOFTMAX-PQC | $[0.01, 0.1, 0.01]$ | 0.9 | 1 | 10 | 4 | 2 |
| | DNN | 0.01 | 0.9 | 1 | 10 | 4 | 16 |
| Cliffwalk-PQC | SOFTMAX-PQC | $[0.01, 0.1, 0.1]$ | 0.9 | 1 | 10 | 4 | 2 |
| | DNN | 0.01 | 0.9 | 1 | 10 | 4 | 16 |
| CognitiveRadio | SOFTMAX-PQC | $[0.01, 0.1, 0.1]$ | 0.9 | 1 | 1 | 3 | 2 to 5 |

Table 2: **Hyperparmeters 1/2.** For PQC policies, we choose 3 distinct learning rates $[\alpha_{\boldsymbol{\phi}}, \alpha_{\boldsymbol{w}}, \alpha_{\boldsymbol{\lambda}}]$ for rotation angles $\boldsymbol{\phi}$, observable weights $\boldsymbol{w}$ and scaling parameters $\boldsymbol{\lambda}$, respectively. For SOFTMAX-PQCs, we take a linear annealing schedule for the inverse temperature parameter $\beta$ starting from 1 and ending up in the final $\beta$. The batch size is counted in number of episodes used to evaluate the gradient of the value function. Depth indicates the number of encoding layers $D_{\mathrm{enc}}$ for PQC policies, or the number of hidden layers for a DNN policy. Width corresponds to the number of qubits $n$ on which acts a PQC (also equal to the dimension $d$ of the environment's state space), or the number of units per hidden layer for a DNN.

| Environment | Model | Entang. topology | Train entang. | Observables | Number of params. | Baseline |
|---|---|---|---|---|---|---|
| CartPole-v1 | SOFTMAX-PQC | All-to-all | Yes | $[wZ_0Z_1Z_2Z_3, (-\ldots)]$ | $\{31, 119\}$ | No |
| | RAW-PQC | All-to-all | Yes | $[Z_0Z_1Z_2Z_3, (-\ldots)]$ | $\{30, 118\}$ | No |
| MountainCar-v0 | SOFTMAX-PQC | One-to-one | No | $[w_0Z_0, w_1Z_0Z_1, w_2Z_1]$ | $\{39, 55\}$ | Yes |
| | RAW-PQC | One-to-one | No | $[P_{0,1}, P_2, P_3]$ | $\{36, 52\}$ | Yes |
| Acrobot-v1 | SOFTMAX-PQC | Circular | Yes | $\left[\boldsymbol{w}_i \cdot (Z_0, \ldots, Z_5)^T\right]_{1 \leq i \leq 3}$ | $\{90, 180\}$ | Yes |
| | RAW-PQC | Circular | Yes | $[P_{0..21}, P_{22..42}, P_{43..63}]$ | $\{72, 162\}$ | Yes |
| SL-PQC | SOFTMAX-PQC | One-to-one | No | $[wZ_0Z_1, (-\ldots)]$ | 37 | No |
| | DNN | ✗ | ✗ | ✗ | 902 | No |
| Cliffwalk-PQC | SOFTMAX-PQC | One-to-one | No | $[wZ_0Z_1, (-\ldots)]$ | 37 | No |
| | DNN | ✗ | ✗ | ✗ | 902 | No |
| CognitiveRadio | SOFTMAX-PQC | Circular | No | $[w_0Z_0, w_1Z_1, \ldots, w_nZ_n]$ | 30 to 75 | No |

Table 3: **Hyperparmeters 2/2.** We call entangling layer a layer of 2-qubit gates in the PQC. Circular and all-to-all topologies of entangling layers are equivalent for $n = 2$ qubits, so we call them one-to-one in that case. When trained, entangling layers are composed of $R_{zz} = e^{-i\theta(Z \otimes Z)/2}$ rotations, otherwise, they are composed of Ctrl-$Z$ gates. For policies with 2 actions, the same observable, up to a sign change, is used for both actions. $Z_i$ refers to a Pauli-$Z$ observable acting on qubit $i$, while $P_{i..j}$ indicates a projection on basis states $i$ to $j$. In the experiments of Sec. 3.2, when the weights of the SOFTMAX-PQC are kept fixed, the observables used for MountainCar-v0 and Acrobot-v1 are $[Z_0, Z_0Z_1, Z_1]$, and those used for CartPole-v1 are $[Z_0Z_1Z_2Z_3, -Z_0Z_1Z_2Z_3]$. The different number of parameters in a given row correspond to the different depths in that same row in Table 2.

## E.2 Shape of the policies learned by PQCs v.s. DNNs

**In CartPole-v1** The results of the Sec. 3 demonstrate that our PQC policies can be trained to good performance in benchmarking environments. To get a feel of the solutions found by our agents, we compare the SOFTMAX-PQC policies learned on CartPole to those learned by standard DNNs (with a softmax output layer), which are known to easily learn close-to-optimal behavior on this task. More specifically, we look at the functions learned by these two models, prior to the application of the softmax normalization function (see Eq. (3)). Typical instances of these functions are depicted in Figure 8. We observe that, while DNNs learn simple, close to piece-wise linear functions of their input state space, PQCs tend to naturally learn very oscillating functions that are more prone to instability. While the results of Schuld *et al.* [11] already indicated that these highly oscillating functions would be natural for PQCs, it is noteworthy to see that these are also the type of functions naturally learned in a direct-policy RL scenario. Moreover, our enhancements to standard PQC classifiers show how to make these highly oscillating functions more amenable to real-world tasks.

**In PQC-generated environments** Fig. 9 shows the analog results to Fig. 5 in the main text but with two different random initializations of the environment-generating PQC. Both confirm our observations. In Fig. 10, we compare the policies learned by prototypical SOFTMAX-PQC and DNN agents in these PQC-generated environments. We observe that the typical policies learned by DNNs are rather simple, with up to 2 (or 3) regions, delimited by close-to-linear boundaries, as opposed to the policies learned by SOFTMAX-PQCs, which delimit red from blue regions with wide margins. These observations highlight the inherent flexibility of SOFTMAX-PQC policies and their suitability to these PQC-generated environments, as opposed to the DNN (and RAW-PQC) policies we consider.

## E.3 Additional numerical simulation on the CognitiveRadio environment

In a related work on value-based RL with PQCs, the authors of Ref. [5] introduced the CognitiveRadio environment as a benchmark to test their RL agents. In this environment, the agent is presented at each interaction step with a binary vector $(0, 0, 0, 1, 0)$ of size $n$ that describes the occupation of $n$ radio channels. Given this state, the agent must select one of the $n$ channels as its communication channel, such as to avoid collision with occupied channels (a $\pm 1$ reward reflects these collisions). The authors of Ref. [5] consider a setting where, in any given state, only one channel is occupied, and its assignment changes periodically over time steps, for an episode length of $100$ steps. While this constitutes a fairly simple task environment with discrete state and action spaces, it allows to test the performance of PQC agents on a family of environments described by their system size $n$ and make claims on the parameter complexity of the PQCs as a function of $n$. As to reproduce the findings of Ref. [5] in a policy-gradient setting, we test the performance of our SOFTMAX-PQC agents on this environment. We find numerically (see Fig. 7) that these achieve a very similar performance to the PQC agents of Ref. [5] on the same system sizes they consider ($n = 2$ to $5$), using PQCs with the same scaling of number of parameters, i.e., $\mathcal{O}(n)$.

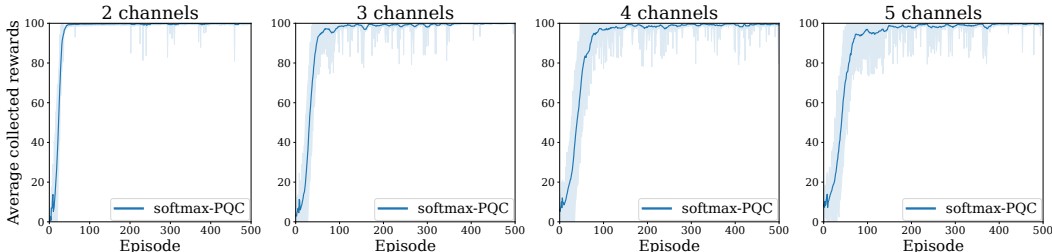

Figure 7: **Performance of our SOFTMAX-PQC agents on the CognitiveRadio environment proposed in Ref. [5].** Average performance of $20$ agents for system sizes (and number of qubits) $n = 2$ to $5$.

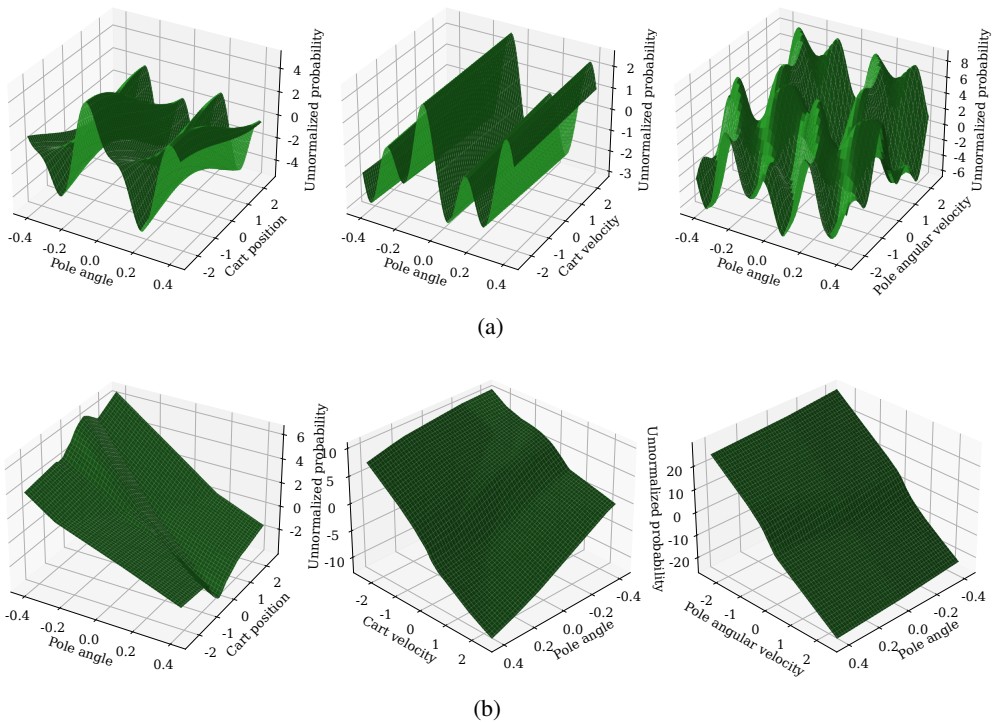

Figure 8: **Prototypical unnormalized policies learned by SOFTMAX-PQC agents and DNN agents in CartPole.** Due to the 4 dimensions of the state space in CartPole, we represent the unnormalized policies learned by (a) SOFTMAX-PQC agents and (b) DNN agents on 3 subspaces of the state space by fixing unrepresented dimensions to 0 in each plot. To get the probability of the agent pushing the cart to the left, one should apply the logistic function (i.e., 2-dimensional softmax) $1/(1 + exp(-z))$ to the $z$-axis values of each plot.

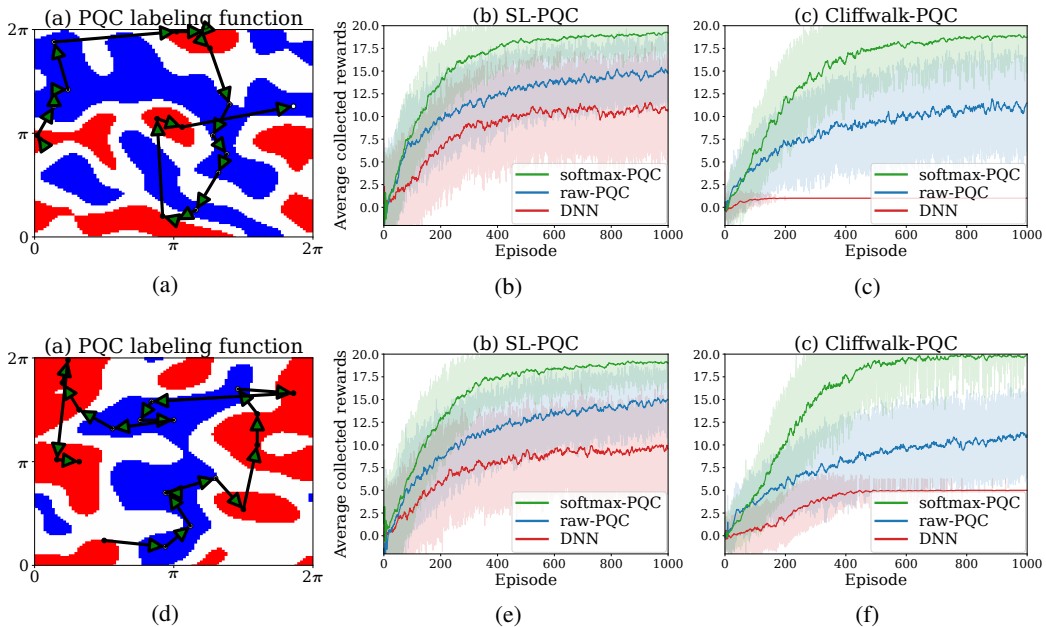

**Figure 9: Different random initializations of PQC-generated environments and their associated learning curves.** See Fig. 5 for details. The additional learning curves (20 agents per curve) of randomly-initialized RAW-PQC agents highlight the hardness of these environments for PQC policies drawn from the same family as the environment-generating PQCs.

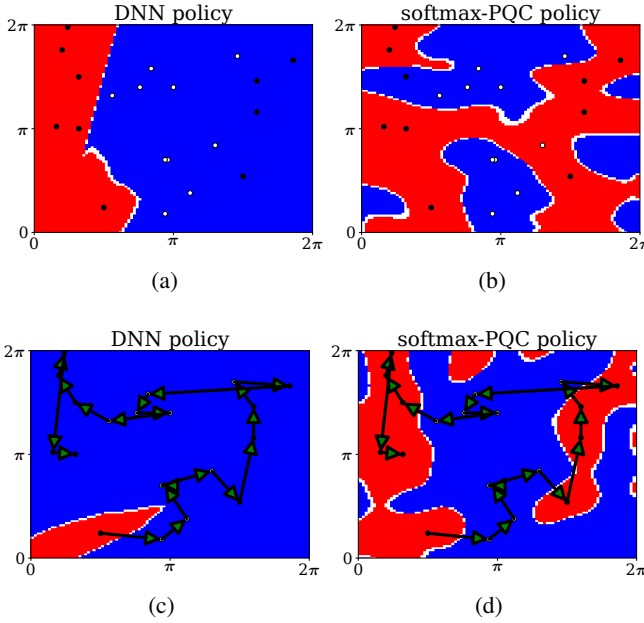

**Figure 10: Prototypical policies learned by SOFTMAX-PQC agents and DNN agents in PQC-generated environments.** All policies are associated to the labeling function of Fig. 9.d. Policies (a) and (b) are learned in the SL-PQC environment while policies (c) and (d) are learned in the Cliffwalk-PQC environment.

## F Supervised learning task of Liu *et al.*

Define $p$ a large prime number, $n = \lceil \log_2(p-1) \rceil$, and $g$ a generator of $\mathbb{Z}_p^* = \{1, 2, \ldots, p-1\}$ (i.e., a $g \in \mathbb{Z}_p^*$ such that $\{g^y, y \in \mathbb{Z}_{p-1}\} = \mathbb{Z}_p^*$). The DLP consists in computing $\log_g x$ on input $x \in \mathbb{Z}_p^*$. Based on DLP, Liu *et al.* [12] define a concept class $\mathcal{C} = \{f_s\}_{s \in \mathbb{Z}_{p-1}}$ over the input space $\mathcal{X} = \mathbb{Z}_p^*$, where each labeling function of this concept class is defined as follows:

$$f_s(x) = \begin{cases} +1, & \text{if } \log_g x \in [s, s + \frac{p-3}{2}], \\ -1, & \text{otherwise.} \end{cases} \tag{9}$$

Each function $f_s : \mathbb{Z}_p^* \to \{-1, 1\}$ hence labels half the elements in $\mathbb{Z}_p^*$ with a label $+1$ and the other half with a label $-1$. We refer to Figure 1 in Ref. [12] for a good visualization of all these objects. The performance of a classifier $f$ is measured in terms of its testing accuracy

$$\text{Acc}_f(f_s) = \Pr_{x \sim \mathcal{X}}[f(x) = f_s(x)].$$

## G Proof of Theorem 1

In the following, we provide constructions of *a)* fully random, *b)* partially random and *c)* fully deterministic environments satisfying the properties of Theorem 1. We consider the three families of environments separately and provide individual lemmas specifying their exact separation properties.

**Fully random: the SL-DLP environment.** This result is near-trivially obtained by noting that any classification problem can be easily mapped to a (degenerate) RL problem. For this, the environment will be an MDP defined as follows: its state space is the input space of the classification problem, its action space comprises all possible labels, rewards are trivially $+1$ for assigning a correct label to an input state and $-1$ otherwise, and the initial and next-state transition probabilities are state-independent and equal to the input distribution of the classification task. The optimal policy of this MDP is clearly the optimal classifier of the corresponding SL task. Consider now the classification task of Liu *et al.*, defined in detail in Appendix F: the input distribution is taken to be uniform on the state space, i.e., $P(s_t) = \frac{1}{|S|}$, and the performance of a classifier $f$ w.r.t. a labeling (or ground truth) function $f^*$ is measured in terms of a testing accuracy

$$\text{Acc}_f(f^*) = \frac{1}{|S|} \sum_s \Pr[f(s) = f^*(s)]. \tag{10}$$

For the MDP associated to this classification task and length-1 episodes of interaction, the value function of any policy $\pi(a|s)$ is given by

$$V_\pi(s_0) = \frac{1}{|S|} \sum_{s_0} (\pi(f^*(s_0)|s_0) - \pi(-f^*(s_0)|s_0))$$

$$= \frac{1}{|S|} \sum_{s_0} 2\pi(f^*(s_0)|s_0) - 1$$

$$= 2\text{Acc}_\pi(f^*) - 1,$$

which is trivially related to the testing accuracy of this policy on the classification task. Note that we also have $V_{\text{rand}}(s_0) = 0$ and $V_{\text{opt}}(s_0) = 1$.

Since these observations hold irrespectively of the labeling function $f^*$, we can show the following result:

**Lemma 4** (Quantum advantage in SL-DLP). *There exists a uniform family of SL-DLP MDPs, each derived from a labeling function $f^*$ of the DLP concept class $\mathcal{C}$ (see Appendix F), for which classical hardness and quantum learnability holds. More specifically, the performance of any classical learner is upper bounded by $1/\text{poly}(n)$, while that of a class of quantum agents is lower bounded by $0.98$ with probability above $2/3$ (over the randomness of their interaction with the environment and noise in their implementation).*

*Proof.* Classical hardness is trivially obtained by contraposition: assuming no classical polynomial-time algorithm can solve DLP, then using Theorem 1 of Liu *et al.*, any classical policy would have

testing accuracy $\text{Acc}_\pi(f^*) \leq 1/2 + 1/\text{poly}(n)$, and hence its value function would be $V_\pi(s_0) \leq 1/\text{poly}(n)$.

For quantum learnability, we define an agent that first collects $\text{poly}(n)$ random length-1 interactions (i.e., a random state $s_0$ and its associated reward for an action $+1$, from which the label $f^*(s_0)$ can be inferred), and use Theorem 2 of Liu *et al.* to train a classifier that has test accuracy at least $0.99$ with probability at least $2/3$ (this process can be repeated $\mathcal{O}\left(\log\left(\delta^{-1}\right)\right)$ times to increase this probability to $1 - \delta$ via majority voting). This classifier has a value function $V_\pi(s_0) \geq 0.98$. $\qquad\square$

Note that this proof trivially generalizes to episodes of interaction with length greater than 1, when preserving the absence of temporal correlation in the states experienced by the agents. For episodes of length $H$, the only change is that the value function of any policy, and hence the bounds we achieve, get multiplied by a factor of $\frac{1-\gamma^H}{1-\gamma}$ for a discount factor $\gamma < 1$ and by a factor $H$ for $\gamma = 1$.

**Partially random: the Cliffwalk-DLP environment.** One major criticism to the result of Lemma 4 is that it applies to a very degenerate, fully random RL environment. In the following, we show that similar results can be obtained in environments based on the same classification problem, but while imposing more temporal structure and less randomness (such constructions were introduced in Ref. [13], but for the purpose of query separations between RL and QRL). For instance, one can consider cliffwalk-type environments, inspired by the textbook "cliff walking" environment of Sutton & Barto [14]. This class of environments differs from the previous SL-DLP environments in its state and reward structure: in any episode of interaction, experienced states follow a fixed "path" structure (that of the cliff) for correct actions, and a wrong action yields to immediate "death" (negative reward and episode termination). We slightly modify this environment to a "slippery scenario" in which, with a $\delta$ probability, any action may lead to a uniformly random position on the cliff. This additional randomness allows us to prove the following separation:

**Lemma 5** (Quantum advantage in Cliffwalk-DLP). *There exists a uniform family of Cliffwalk-DLP MDPs with arbitrary slipping probability $\delta \in [0.86, 1]$ and discount factor $\gamma \in [0, 0.9]$, each derived from a labeling function $f^*$ of the DLP concept class $\mathcal{C}$, for which classical hardness and quantum learnability holds. More specifically, the performance of any classical learner is upper bounded by $V_{\text{rand}}(s_0) + 0.1$, while that of a class of quantum agents is lower bounded by $V_{\text{opt}}(s_0) - 0.1$ with probability above $2/3$ (over the randomness of their interaction with the environment and noise in their implementation). Since $V_{\text{rand}}(s_0) \leq -\frac{1}{2}$ and $V_{\text{opt}} = 0$, we always have a classical-quantum separation.*

The proof of this lemma is deferred to Appendix H for clarity.

**Fully deterministic: the Deterministic-DLP environment.** The simplest example of a deterministic RL environment where separation can be proven is a partially observable MDP (POMDP) defined as follows: it constitutes a 1-D chain of states of length $k + 2$, where $k$ is $\text{poly}(n)$. We refer to the first $k$ states as "training states", and we call the last two states "test" and "limbo" states, respectively. The training states are of the form $(x, f_s(x))$, i.e., a point uniformly sampled and its label. The actions are $+1, -1$, and both lead to the same subsequent state on the chain (since the same $(x, f_s(x))$ can appear twice in the chain, this is the reason why the environment is partially observable), and no reward is given for the first $k$ states. In the test state, the agent is only given a point $x$ with no label. A correct action provides a reward of 1 and leads to the beginning of the chain, while an incorrect action leads to the limbo state, which self-loops for both actions and has no rewards. In other words, after poly-many examples where the agent can learn the correct labeling, it is tested on one state. Failure means it will never obtain a reward.

For each concept $f_s$, we define exponentially many environments obtained by random choices of the states appearing in the chain. In a given instance, call $T = (x_0, \ldots, x_{k-1})$ the training states of that instance, $x_k$ its testing state and $l$ its limbo state. The interaction of an agent with the environment is divided into episodes of length $k + 1$, but the environment keeps memory of its state between episodes. This means that, while the first episode starts in $x_0$, depending on the performance of the agent, later episodes start either in $x_0$ or in $l$. For a policy $\pi$, we define the value $V_\pi(s_0)$ as the expected reward[1]

---

[1]Note that we assume here a discount factor $\gamma = 1$, but our results would also hold for an arbitrary $\gamma > 0$, if we scale the reward of the testing state to $\gamma^{-k}$.

of this policy in any episode of length $k + 1$ with an initial state $s_0 \in \{x_0, l\}$. Since the testing state $x_k$ is the only state to be rewarded, we can already note that $V_\pi(x_0) = \pi(f^*(x_k)|T, x_k)$, that is, the probability of the policy correctly labeling the testing state $x_k$ after having experienced the training states $T$. Also, since $s_0 \in \{x_0, l\}$ and $V_\pi(l) = 0$, we have $V_\pi(x_0) \geq V_\pi(s_0)$.

With this construction, we obtain the following result:

**Lemma 6** (Quantum advantage in Deterministic-DLP)**.** *There exists a uniform family of Deterministic-DLP POMDPs (exponentially many instances for a given concept $f_s$ of the DLP classification problem) where:*
*1) (classical hardness) if there exists a classical learning agent which, when placed in a randomly chosen instance of the environment, has value $V_c(s_0) \geq 1/2 + 1/\mathrm{poly}(n)$ (that is, $1/\mathrm{poly}(n)$ better than a random agent), with probability at least $0.845$ over the choice of environment and the randomness of its learning algorithm, then there exists an efficient classical algorithm to solve DLP,*
*2) (quantum learnability) there exists a class of quantum agents that attains a value $V_q(s_0) = 1$ (that is, the optimal value) with probability at least $0.98$ over the choice of environment and randomness of the learning algorithm.*

The proof of this lemma is deferred to Appendix I for clarity.

By combining our three lemmas, and taking the weakest separation claim for the cases *ii)* and *iii)*, we get Theorem 1. For the interested reader, we list the following remarks, relating to the proofs of these lemmas:

- SL-DLP and Deterministic-DLP are the two closest environments to the DLP classification task of Liu *et al.* While the value function in SL-DLP is trivially equivalent to the accuracy of the classification problem, we find the value function in Deterministic-DLP to be *weaker* than this accuracy. Namely, a high accuracy trivially leads to a high value while a high (or non-trivial) value does not necessarily lead to a high (or non-trivial) accuracy (in all these cases, the high probability over the randomness of choosing the environments and of the learning algorithms is implied). This explains why the classical hardness statement for Deterministic-DLP is weaker than in SL-DLP.

- In Cliffwalk-DLP, it is less straightforward to relate the testing accuracy of a policy to its performance on the deterministic parts of the environment, which explains why we trivially upper bound this performance by 0 on these parts. We believe however that these deterministic parts will actually make the learning task much harder, since they strongly restrict the part of the state space the agents can see. This claim is supported by our numerical experiments in Sec. 4.2. Also, since we showed classical hardness for fully deterministic environments, it would be simple to construct a variant of Cliffwalk-DLP where these deterministic parts would be provably hard as well.

# H  Proof of Lemma 5

Consider a slippery cliffwalk environment defined by a labeling function $f^*$ in the concept class $\mathcal{C}$ of Liu *et al.* This cliffwalk has $p - 1$ states ordered, w.l.o.g., in their natural order, and correct actions (the ones that do not lead to immediate "death") $f^*(i)$ for each state $i \in \mathbb{Z}_p^*$. For simplicity of our proofs, we also consider circular boundary conditions (i.e, doing the correct action on the state $p - 1$ of the cliff leads to the state 1), random slipping at each interaction step to a uniformly sampled state on the cliff with probability $\delta > 0$, an initialization of each episode in a uniformly sampled state $i \in \mathbb{Z}_p^*$, and a $0$ $(-1)$ reward for doing the correct (wrong) action in any given state.

## H.1  Upper bound on the value function

The value function of any policy $\pi$ which has probability $\pi(i)$ (we abbreviate $\pi(f^*(i)|i)$ to $\pi(i)$) of doing the correct action in state $i \in \mathbb{Z}_p^*$ is given by:

$$V_\pi(i) = \pi(i)\gamma \left( (1 - \delta)V_\pi(i + 1) + \delta \frac{1}{p - 1} \sum_{j=1}^{p-1} V_\pi(j) \right) - (1 - \pi(i)) \qquad (11)$$

Since this environment only has negative rewards, we have that $V_\pi(i) \leq 0$ for any state $i$ and policy $\pi$, which allows us to write the following inequality:

$$V_\pi(i) \leq \pi(i)\gamma \left( \delta \frac{1}{p-1} \sum_{j=1}^{p-1} V_\pi(j) \right) - (1 - \pi(i))$$

We use this inequality to bound the following term:

$$\frac{1}{p-1} \sum_{i=1}^{p-1} V_\pi(i) \leq \frac{1}{p-1} \sum_{i=1}^{p-1} \left( \pi(i) \frac{\gamma\delta}{p-1} \sum_{j=1}^{p-1} V_\pi(j) - (1 - \pi(i)) \right)$$

$$= \left( \frac{1}{p-1} \sum_{i=1}^{p-1} \pi(i) \right) \left( \frac{\gamma\delta}{p-1} \sum_{j=1}^{p-1} V_\pi(j) + 1 \right) - 1$$

We note that the first factor is exactly the accuracy of the policy $\pi$ on the classification task of Liu *et al.*:

$$\mathrm{Acc}_\pi(f^*) = \frac{1}{p-1} \sum_{i=1}^{p-1} \pi(i).$$

We hence have:

$$\frac{1}{p-1} \sum_{i=1}^{p-1} V_\pi(i) \leq \mathrm{Acc}_\pi(f^*) \left( \gamma\delta \frac{1}{p-1} \sum_{j=1}^{p-1} V_\pi(j) + 1 \right) - 1$$

which is equivalent to:

$$\frac{1}{p-1} \sum_{i=1}^{p-1} V_\pi(i) \leq \frac{\mathrm{Acc}_\pi(f^*) - 1}{1 - \mathrm{Acc}_\pi(f^*)\gamma\delta}$$

when $\mathrm{Acc}_\pi(f^*)\gamma\delta < 1$.
We now note that this average value function is exactly the value function evaluated on the initial state $s_0$ of the agent, since this state is uniformly sampled from $\mathbb{Z}_p^*$ for every episode. Hence,

$$V_\pi(s_0) \leq \frac{\mathrm{Acc}_\pi(f^*) - 1}{1 - \mathrm{Acc}_\pi(f^*)\gamma\delta} \tag{12}$$

### H.2 Lower bound on the value function

Again, by noting in Eq. (11) that we have $V_\pi(i) \leq 0$ and $\pi(i) \leq 1$ for any policy $\pi$ and state $i \in \mathbb{Z}_p^*$, we have:

$$V_\pi(i) \geq \gamma \left( (1-\delta)V_\pi(i+1) + \frac{\delta}{p-1} \sum_{j=1}^{p-1} V_\pi(j) \right) - (1 - \pi(i))$$

We use this inequality to bound the value function at the initial state $s_0$:

$$V_\pi(s_0) = \frac{1}{p-1} \sum_{i=1}^{p-1} V_\pi(i)$$

$$\geq \gamma \left( \frac{1-\delta}{p-1} \sum_{i=1}^{p-1} V_\pi(i+1) + \frac{\delta}{p-1} \sum_{j=1}^{p-1} V_\pi(j) \right) + \frac{1}{p-1} \sum_{i=1}^{p-1} \pi(i) - 1$$

$$= \gamma \left( (1-\delta)V_\pi(s_0) + \delta V_\pi(s_0) \right) + \mathrm{Acc}_\pi(f^*) - 1$$

$$= \gamma V_\pi(s_0) + \mathrm{Acc}_\pi(f^*) - 1$$

by using the circular boundary conditions of the cliffwalk in the third line.
This inequality is equivalent to:

$$V_\pi(s_0) \geq \frac{\mathrm{Acc}_\pi(f^*) - 1}{1 - \gamma} \tag{13}$$

when $\gamma < 1$.

## H.3 Bounds for classical hardness and quantum learnability

We use the bounds derived in the two previous sections to prove classical hardness and quantum learnability of this task environment, as stated in Lemma 5.

For this, we start by noting the following expression for the value function of a random policy (one that does random actions in all states):

$$V_{\text{rand}}(s_0) = \frac{\gamma}{2}\left(\frac{1-\delta}{p-1}\sum_{i=1}^{p-1}V_{\text{rand}}(i+1) + \frac{\delta}{p-1}\sum_{j=1}^{p-1}V_{\text{rand}}(j)\right) - \frac{1}{2}$$

$$= \frac{\gamma}{2}V_{\text{rand}}(s_0) - \frac{1}{2} = -\frac{1}{2-\gamma}$$

again due to the circular boundary conditions of the cliffwalk and the resulting absence of termination conditions outside of "death".
As for the value function of the optimal policy, this is trivially $V_{\text{opt}} = 0$.

### H.3.1 Proof of classical hardness

For any policy $\pi$, we define the function $g(x, \delta, \gamma) = V(x, \delta, \gamma) - V_{\text{rand}}(\gamma)$, where we adopt the short-hand notation $x = \text{Acc}_\pi(f^*)$ and call $V$ the upper bound on the value function $V_\pi(s_0)$ of $\pi$. The expression of $g(x, \delta, \gamma)$ (for $(x, \delta, \gamma) \neq (1, 1, 1)$) is given by:

$$g(x, \delta, \gamma) = \frac{x-1}{1-\delta\gamma x} + \frac{1}{2-\gamma} \tag{14}$$

To prove classical hardness, it is sufficient to show that $x \leq 0.51$ implies $g(x, \delta, \gamma) \leq 0.1$ for $\delta \in [\delta_0, 1]$, $\gamma \in [0, \gamma_1]$ and a $\{\delta_0, \gamma_1\}$ pair of our choosing. To see this, notice that the contraposition gives $x = \text{Acc}_\pi(f^*) > 0.51$ which is sufficient to construct an efficient algorithm that solves DLP. To achieve this result, we show the three following inequalities, $\forall\, x \leq 0.51$ and $\forall\, (\delta, \gamma) \in [\delta_0, 1] \times [0, \gamma_1]$:

$$g(x, \delta, \gamma) \overset{(i)}{\leq} g(0.51, \delta, \gamma) \overset{(ii)}{\leq} g(0.51, \delta_0, \gamma) \overset{(iii)}{\leq} g(0.51, \delta_0, \gamma_1)$$

where $\delta_0$ and $\gamma_1$ are chosen such that $g(0.51, \delta_0, \gamma_1) \leq 0.1$.

*Proof of (i).* We look at the derivative of $g$ w.r.t. $x$:

$$\frac{\partial g(x, \delta, \gamma)}{\partial x} = \frac{1-\delta\gamma}{(1-\delta\gamma x)^2} \geq 0 \quad \forall (x, \delta, \gamma) \in [0, 1]^3 \backslash (1, 1, 1)$$

and hence $g$ is an increasing function of $x$, which gives our inequality. $\qquad\square$

*Proof of (ii).* We look at the derivative of $g$ w.r.t. $\delta$:

$$\frac{\partial g(x, \delta, \gamma)}{\partial \delta} = \frac{\gamma(x-1)x}{(1-\delta\gamma x)^2} \leq 0 \quad \forall (x, \delta, \gamma) \in [0, 1]^3 \backslash (1, 1, 1)$$

and hence $g$ is a decreasing function of $\delta$, which gives our inequality. $\qquad\square$

*Proof of (iii).* We look at the derivative of $g$ w.r.t. $\gamma$:

$$\frac{\partial g(x, \delta, \gamma)}{\partial \gamma} = \frac{\delta(x-1)x}{(1-\delta\gamma x)^2} + \frac{1}{(2-\gamma)^2} \quad \forall (x, \delta, \gamma) \in [0, 1]^3 \backslash (1, 1, 1)$$

We have:

$$\frac{\partial g(x, \delta, \gamma)}{\partial \gamma} \geq 0 \Leftrightarrow \left((\delta x)^2 + \delta(x^2 - x)\right)\gamma^2 - 2\delta(2x^2 - x)\gamma + 4\delta(x^2 - x) + 1 \geq 0$$

By setting $x = 0.51$ and $\delta = 0.86$, we find

$$\frac{\partial g(0.51, 0.86, \gamma)}{\partial \gamma} \geq 0 \quad \forall \gamma \in [0, 1]$$

since the roots of the second-degree polynomial above are approximately $\{-2.91, 2.14\}$ and we have $(\delta x)^2 + \delta(x-1)x \approx -0.0225 < 0$.
Hence $g(0.51, \delta_0, \gamma)$ is an increasing function of $\gamma$, which gives our inequality. $\qquad\square$

Given that $g(0.51, 0.86, 0.9) \approx 0.0995 < 0.1$, we then get our desired result for $\delta_0 = 0.86$ and $\gamma_1 = 0.9$. Noting that $V_\pi(s_0) - V_{\mathrm{rand}}(\gamma) \leq g(x, \delta, \gamma) \leq 0.1$ from Eq. (12), we hence have classical hardness $\forall \, (\delta, \gamma) \in [\delta_0, 1] \times [0, \gamma_1]$.

### H.3.2 Proof of quantum learnability

Proving quantum learnability is more trivial, since, for $\mathrm{Acc}_\pi(f^*) \geq 0.99$ and $\gamma \leq 0.9$, we directly have, using Eq. (13):

$$V_\pi(s_0) \geq -0.1 = V_{\mathrm{opt}} - 0.1$$

To conclude this proof, we still need to show that we can obtain in this environment a policy $\pi$ such that $\mathrm{Acc}_\pi(f^*) \geq 0.99$ with high probability. For that, we use agents that first collect $\mathrm{poly}(n)$ *distinct* samples (states $s$ and their inferred labels $f^*(s)$) from the environment (distinct in order to avoid biasing the distribution of the dataset with the cliffwalk temporal structure). This can be done efficiently in $\mathrm{poly}(n)$ interactions with the environment, since each episode is initialized in a random state $s_0 \in \mathbb{Z}_p^*$. We then use the learning algorithm of Liu *et al.* to train a classifier $\pi$ with the desired accuracy, with high probability.

# I  Proof of Lemma 6

## I.1  Proof of classical hardness

Suppose that a polynomial-time classical agent achieves a value $V_c(s_0) \geq \frac{1}{2} + \frac{1}{\mathrm{poly}(n)}$ with probability $(1-\delta)$ over the choice of environment and the randomness of its learning algorithm. We call "success" the event $V_c(s_0) \geq \frac{1}{2} + \frac{1}{\mathrm{poly}(n)}$ and $S_\delta$ the subset of the instances $S = \{T, x_k\}$ for which, theoretically, a run of the agent would "succeed" (this is hence a set that depends on the randomness of the agent).

Note that, on every instance in $S_\delta$, $\pi(f^*(x_k)|T, x_k) = V_c(x_0) \geq V_c(s_0) \geq \frac{1}{2} + \frac{1}{\mathrm{poly}(n)}$. Since this probability is bounded away from $1/2$ by an inverse polynomial, this means that we can "boost" it to a larger probability $(1-\varepsilon)$. More specifically, out of the policy $\pi$ obtained after interacting for $k$ steps with the environment, we define a classifier $f_c$ acting on $x_k$ such that we sample $\mathcal{O}\left(\log\left(\varepsilon^{-1}\right)\right)$-many times from $\pi(a|T, x_k)$ and label $x_k$ by majority vote. For the instances in $S_\delta$, the probability of correctly labeling $x_k$ is $\Pr\left[f_c(x_k) = f^*(x_k)\right] \geq 1 - \varepsilon$.

Define $P(T) = \Pr[\mathrm{T} = T]$ and $P(x_k) = \Pr[\mathrm{x_k} = x_k]$ the probabilities of sampling certain training states $T$ and a testing state $x_k$, when choosing an instance of the environment. We now look at the following quantity:

$$
\begin{aligned}
\mathbb{E}_{P(T)}\left[\mathrm{Acc}_{f_c}(T)\right] &= \sum_T P(T) \sum_{x_k} P(x_k) \Pr\left[f_c(x_k) = f^*(x_k)|T, x_k\right] \\
&= \sum_{T, x_k} P(T, x_k) \Pr\left[f_c(x_k) = f^*(x_k)|T, x_k\right] \\
&\geq \sum_{T, x_k} P(T, x_k) \Pr\left[\mathrm{success}|T, x_k\right] \times \Pr\left[f_c(x_k) = f^*(x_k)|T, x_k, \mathrm{success}\right] \\
&\geq (1-\delta)(1-\varepsilon)
\end{aligned}
$$

since $\Pr\left[f_c(x_k) = f^*(x_k)|T, x_k\right] \geq 1 - \varepsilon$ for instances in $S_\delta$ and $\sum_{T, x_k} P(T, x_k) \Pr\left[\mathrm{success}|T, x_k\right] \geq 1 - \delta$ by definition.

In the following, we set $1 - \varepsilon = 0.999$ and $1 - \delta \geq 0.845$ (the reason for this becomes apparent below), such that:

$$\mathbb{E}_{P(T)}\left[\mathrm{Acc}_{f_c}(T)\right] \geq 0.844155 > \frac{5}{6} + \frac{1}{96} \tag{15}$$

Now, consider the following learning algorithm: given a training set $T$, construct a Deterministic-DLP environment that uses this $T$ and a randomly chosen $x_k$, and define the classifier $f_c$ that boosts the $\pi(a|T, x_k)$ obtained by running our classical agent on this environment (as explained above). We want to show that $f_c$ has accuracy $\mathrm{Acc}_{f_c}(T) \geq \frac{1}{2} + \frac{1}{\mathrm{poly}(n)}$ with probability at least $2/3$ over the choice of $T$ and the randomness of its construction, which is sufficient to solve DLP classically. For that, we show a stronger statement. Call $\mathcal{T}_{\mathrm{succ}}$ the subset of all instances of training states $\mathcal{T} = \{T\}$

for which $\mathrm{Acc}_{f_c}(T) \geq \frac{1}{2} + \frac{1}{\mathrm{poly}(n)}$. We prove by contradiction that $|\mathcal{T}_{\mathrm{succ}}| \geq \frac{2|\mathcal{T}|}{3}$:

Assume $|\mathcal{T}_{\mathrm{succ}}| < \frac{2|\mathcal{T}|}{3}$, then

$$\mathbb{E}_{P(T)}\left[\mathrm{Acc}_{f_c}(T)\right] = \sum_T P(T)\mathrm{Acc}_{f_c}(T)$$

$$= \frac{1}{|\mathcal{T}|}\left(\sum_{T \in \mathcal{T}_{\mathrm{succ}}} \mathrm{Acc}_{f_c}(T) + \sum_{T \notin \mathcal{T}_{\mathrm{succ}}} \mathrm{Acc}_{f_c}(T)\right)$$

$$< \frac{|\mathcal{T}_{\mathrm{succ}}|}{|\mathcal{T}|} \times 1 + \frac{|\mathcal{T}| - |\mathcal{T}_{\mathrm{succ}}|}{|\mathcal{T}|}\left(\frac{1}{2} + \frac{1}{\mathrm{poly}(n)}\right)$$

$$< \frac{5}{6} + \frac{1}{3\mathrm{poly}(n)} < 0.844155$$

for large enough $n$, in contradiction with Eq. (15).

Hence, with probability at least $2/3$ over the choice of training states and the randomness of the learning algorithm, our constructed classifier has accuracy $\mathrm{Acc}_{f_c}(T) \geq \frac{1}{2} + \frac{1}{\mathrm{poly}(n)}$. By using Theorem 8, Remark 1 of Liu *et al.*, this is sufficient to construct an efficient classical algorithm that solves DLP.

### I.2 Proof of quantum learnability

Using the learning algorithm of Liu *et al.*, we can construct a quantum classifier that achieves accuracy $\mathrm{Acc}_q(T) \geq 0.99$ with probability at least $2/3$ over the randomness of the learning algorithm and the choice of training states $T$, of length $|T| = \mathrm{poly}(n)$. Now define instead training states $T$ of length $|T| = M\mathrm{poly}(n)$, for $M = \mathcal{O}\left(\log\left(\delta'^{-1}\right)\right)$ (hence $|T|$ is still polynomial in $n$), and use each of the $M$ segments of $T$ to train $M$ independent quantum classifiers. Define $f_q$ as a classifier that labels $x_k$ using a majority vote on the labels assigned by each of these classifiers. This constructed classifier has accuracy $\mathrm{Acc}_{f_q}(T) \geq 0.99$ with now probability $1 - \delta'$ over the choice of training states $T$ and the randomness of the learning algorithm.

We then note that, by calling "success" the event $\mathrm{Acc}_{f_q}(T) \geq 0.99$, we have:

$$\sum_{T, x_k} P(T, x_k)\mathrm{Pr}\left[V_q(x_0) = 1 | T, x_k\right]$$

$$\geq \sum_T P(T) \sum_{x_k} P(x_k)\mathrm{Pr}\left[\mathrm{success}|T\right] \times \mathrm{Pr}\left[V_q(x_0) = 1 | T, x_k, \mathrm{success}\right]$$

$$= \sum_T P(T)\mathrm{Pr}\left[\mathrm{success}|T\right] \sum_{x_k} P(x_k) \times \mathrm{Pr}\left[f_q(x_k) = f^*(x_k) | T, x_k, \mathrm{success}\right]$$

$$= \sum_T P(T)\mathrm{Pr}\left[\mathrm{success}|T\right]\mathrm{Acc}_{f_q}(T)$$

$$\geq (1 - \delta') \times 0.99$$

which means that our constructed agent achieves a value $V_q(x_0) = 1$ (which also implies $V_q(s_0) = 1$) with probability at least $(1 - \delta') \times 0.99$ over the choice of environment and the randomness of the learning algorithm. By setting $(1 - \delta') = 0.98/0.99$, we get our statement.

## J    Construction of a PQC agent for the DLP environments

In the two following appendices, we construct a PQC classifier that can achieve close-to-optimal accuracy in the classification task of Liu *et al.* [12] (see Appendix F), and can hence also be used as a learning model in the DLP environments defined in Sec. 4.1.

### J.1    Implicit v.s. explicit quantum SVMs

To understand the distinction between the quantum learners of Liu *et al.* and the PQC policies we are constructing here, we remind the reader of the two models for quantum SVMs defined in Ref. [15]:

the explicit and the implicit model. Both models share a feature-encoding unitary $U(x)$ that encodes data points $x$ into feature state $|\phi(x)\rangle = U(x) |0^{\otimes n}\rangle$.

In the implicit model, one first evaluates the kernel values

$$K(x_i, x_j) = |\langle \phi(x_i)|\phi(x_j)\rangle|^2 \tag{16}$$

for the feature states associated to every pair of data points $\{x_i, x_j\}$ in the dataset, then uses the resulting kernel matrix in a classical SVM algorithm. This algorithm returns a hyperplane classifier in feature space, defined by its normal vector $\langle w| = \sum_i \alpha_i \langle \phi(x_i)|$ and bias $b$, such that the sign of $|\langle w|\phi(x)\rangle|^2 + b$ gives the label of $x$.

In the explicit model, the classifier is instead obtained by training a parametrized $|w_\theta\rangle$. Effectively, this classifier is implemented by applying a variational unitary $V(\theta)$ on the feature states $|\phi(x)\rangle$ and measuring the resulting quantum states using a fixed observable, with expectation value $|\langle w_\theta|\phi(x)\rangle|^2$.

In the following sections, we describe how the implicit quantum SVMs of Liu *et al.* can be transformed into explicit models while guaranteeing that they can still represent all possible optimal policies in the DLP environments. And in Appendix K, we show that, even under similar noise considerations as Liu *et al.*, these optimal policies can also be found using poly$(n)$ random data samples.

## J.2 Description of the PQC classifier

As we just described, our classifier belongs to a family of so-called explicit quantum SVMs. It is hence described by a PQC with two parts: a feature-encoding unitary $U(x)$, which creates features $|\phi(x)\rangle = U(x) |0^{\otimes n}\rangle$ when applied to an all-0 state, followed by a variational circuit $V(\theta)$ parametrized by a vector $\theta$. The resulting quantum state is then used to measure the expectation value $\langle O \rangle_{x,\theta}$ of an observable $O$, to be defined. We rely on the same feature-encoding unitary $U(x)$ as the one used by Liu *et al.*, i.e., the unitary that creates feature states of the form

$$|\phi(x)\rangle = \frac{1}{\sqrt{2^k}} \sum_{i=0}^{2^k-1} |x \cdot g^i\rangle \tag{17}$$

for $k = n - t \log(n)$, where $t$ is a constant defined later, under noise considerations. This feature state can be seen as the uniform superposition of the image (under exponentiation $s' \mapsto g^{s'}$) of an interval of integers $[\log_g(x), \log_g(x) + 2^k - 1]$ in log-space. Note that $U(x)$ can be implemented in $\widetilde{\mathcal{O}}(n^3)$ operations [12].

By noting that every labeling functions $f_s \in \mathcal{C}$ to be learned (see Eq. (9)) is delimiting two equally-sized intervals of $\log(\mathbb{Z}_p^*)$, we can restrict the decision boundaries to be learned by our classifier to be half-space dividing hyperplanes in log-space. In feature space, this is equivalent to learning separating hyperplanes that are normal to quantum states of the form:

$$|\phi_{s'}\rangle = \frac{1}{\sqrt{(p-1)/2}} \sum_{i=0}^{(p-3)/2} \left|g^{s'+i}\right\rangle. \tag{18}$$

Noticeably, for input points $x$ such that $\log_g(x)$ is away from some delimiting regions around $s$ and $s + \frac{p-3}{2}$, we can notice that the inner product $|\langle \phi(x)|\phi_s\rangle|^2$ is either $\Delta = \frac{2^{k+1}}{p-1}$ or 0, whenever $x$ is labeled $+1$ or $-1$ by $f_s$, respectively. This hence leads to a natural classifier to be built, assuming overlaps of the form $|\langle \phi(x)|\phi_{s'}\rangle|^2$ can be measured:

$$h_{s'}(x) = \begin{cases} 1, & \text{if } |\langle \phi(x)|\phi_{s'}\rangle|^2/\Delta \geq 1/2, \\ -1, & \text{otherwise} \end{cases} \tag{19}$$

which has an (ideal) accuracy $1 - \Delta$ whenever $s' = s$.

To complete the construction of our PQC classifier, we should hence design the composition of its variational part $V(\theta)$ and measurement $O$ such that they result in expectation values of the form $\langle O \rangle_{x,\theta} = |\langle \phi(x)|\phi_{s'}\rangle|^2$. To do this, we note that, for $|\phi_{s'}\rangle = \hat{V}(s') |0\rangle$, the following equality holds:

$$|\langle \phi(x)|\phi_{s'}\rangle|^2 = \left|\langle 0^{\otimes n}| \hat{V}(s')^\dagger U(x_i) |0^{\otimes n}\rangle\right|^2$$
$$= \text{Tr}\left[|0^{\otimes n}\rangle \langle 0^{\otimes n}| \rho(x, s')\right]$$

where $\rho(x, s') = |\psi(x, s')\rangle \langle \psi(x, s')|$ is the density matrix of the quantum state $|\psi(x, s')\rangle = \hat{V}(s')^\dagger U(x_i) |0^{\otimes n}\rangle$. Hence, an obvious choice of variational circuit is $V(\boldsymbol{\theta}) = \hat{V}(s')$, combined with a measurement operator $O = |0^{\otimes n}\rangle \langle 0^{\otimes n}|$. Due to the similar nature of $|\phi'_s\rangle$ and $|\phi(x)\rangle$, it is possible to use an implementation for $\hat{V}(s')$ that is similar to that of $U(x_i)$ (take $x_i = g^{s'}$ and $k \approx n/2$).[2] We also note that, for points $x$ such that $\log_g(x)$ is $(p-1)\Delta/2$ away from the boundary regions of $h_{s'}$, the non-zero inner products $|\langle \phi(x)|\phi_{s'}\rangle|^2$ are equal to $\Delta = \mathcal{O}(n^{-t})$. These can hence be estimated efficiently to additive error, which allows to efficiently implement our classifier $h_{s'}$ (Eq. (19)).

## J.3 Noisy classifier

In practice, there will be noise associated with the estimation of the inner products $|\langle \phi(x)|\phi_{s'}\rangle|^2$, namely due to the additive errors associated to sampling. Similarly to Liu *et al.*, we model noise by introducing a random variable $e_{is'}$ for each data point $x_i$ and variational parameter $g^{s'}$, such that the estimated inner product is $|\langle \phi(x_i)|\phi_{s'}\rangle|^2 + e_{is'}$. This random variable satisfies the following equations:

$$\begin{cases} e_{is'} \in [-\Delta, \Delta] \\ \mathbb{E}[e_{is'}] = 0 \\ \mathrm{Var}[e_{is'}] \leq 1/R \end{cases}$$

where $R$ is the number of circuit evaluations used to estimate the inner product. We hence end up with a noisy classifier:

$$\widetilde{h}_{s'}(x_i) = \begin{cases} 1, & \text{if } \left(|\langle \phi(x_i)|\phi_{s'}\rangle|^2 + e_{is'}\right)/\Delta \geq 1/2, \\ -1, & \text{otherwise} \end{cases}$$

The noise has the effect that some points which would be correctly classified by the noiseless classifier have now a non zero probability of being misclassified. To limit the overall decrease in classification accuracy, we focus on limiting the probability of misclassifying points $x_i$ such that $\log_g(x_i)$ is $(p-1)\Delta/2$ away from the boundary points $s'$ and $s' + \frac{p-3}{2}$ of $g_{s'}$. We call $I_{s'}$ the subset of $\mathbb{Z}_p^*$ comprised of these points. For points in $I_{s'}$, the probability of misclassification is that of having $|e_{is'}| \geq \Delta/2$. We can use Chebyshev's inequality to bound this probability:

$$\Pr\left(|e_{is'}| \geq \frac{\Delta}{2}\right) \leq \frac{4}{\Delta^2 R} \tag{20}$$

since $\mathbb{E}[e_{is'}] = 0$ and $\mathrm{Var}[e_{is'}] \leq 1/R$.

## K  Proof of trainability of our PQC agent in the SL-DLP environment

In this Appendix, we describe an optimization algorithm to train the variational parameter $g^{s'}$ of the PQC classifier we defined in Appendix J. This task is non-trivial for three reasons: 1) even by restricting the separating hyperplanes accessible by our classifier, there are still $p-1$ candidates, which makes an exhaustive search for the optimal one intractable; 2) noise in the evaluation of the classifier can potentially heavily perturb its loss landscape, which can shift its global minimum and 3) decrease the testing accuracy of the noisy classifier. Nonetheless, we show that all these considerations can be taken into account for a simple optimization algorithm, such that it returns a classifier with close-to-optimal accuracy with high probability of success. More precisely, we show the following Theorem:

**Theorem 2.** *For a training set of size $n^c$ such that $c \geq \max\left\{\log_n(8/\delta), \log_n\left(\frac{\log(\delta/2)}{\log(1-2n^{-t})}\right)\right\}$ for $t \geq \max\left\{3\log_n(8/\delta), \log_n(16/\varepsilon)\right\}$ in the definition of $\Delta$, and a number of circuit evaluations per inner product $R \geq \max\left\{\frac{4n^{2(t+c)}}{\delta}, \frac{128}{\varepsilon^3}\right\}$, then our optimization algorithm returns a noisy classifier $\widetilde{h}_{s'}$ with testing accuracy $\mathrm{Acc}_{\widetilde{h}_{s'}}(f_s)$ on the DLP classification task of* Liu *et al. such that*

$$\Pr\left(\mathrm{Acc}_{\widetilde{h}_{s'}}(f_s) \geq 1 - \varepsilon\right) \geq 1 - \delta.$$

---

[2]Note that we write $\hat{V}(s')$ and $U_{s'}$ to be parametrized by $s'$ but the true variational parameter here is $g^{s'}$, since we work in input space and not in log-space.

The proof of this Theorem is detailed below.

Given a training set $X \subset \mathcal{X}$ polynomially large in $n$, i.e., $|X| = n^c$, define the training loss:

$$\mathcal{L}(s') = \frac{1}{2|X|} \sum_{x \in X} |h_{s'}(x) - f_s(x)|$$

and its noisy analog:

$$\widetilde{\mathcal{L}}(s') = \frac{1}{2|X|} \sum_{x \in X} \left| \widetilde{h}_{s'}(x) - f_s(x) \right|$$

Our optimization algorithm goes as follows: using the noisy classifier $\widetilde{h}_{s'}$, evaluate the loss function $\widetilde{\mathcal{L}}\left(\log_g(x)\right)$ for each variational parameter $g^{s'} = x \in X$, then set

$$g^{s'} = \operatorname{argmin}_{x \in X} \widetilde{\mathcal{L}}(\log_g(x)).$$

This algorithm is efficient in the size of the training set, since it only requires $|X|^2$ evaluations of $\widetilde{h}_{s'}$. To prove Theorem 2, we show first that we can enforce $\operatorname{argmin}_{x \in X} \widetilde{\mathcal{L}}(\log_g(x)) = \operatorname{argmin}_{x \in X} \mathcal{L}(\log_g(x))$ with high probability (Lemma 7), and second, that this algorithm also leads to $s'$ close to the optimal $s$ in log-space with high probability (Lemma 8).

**Lemma 7.** *For a training set of size $n^c$ such that $c \geq \log_n(8/\delta)$, a $t \geq 3c$ in the definition of $\Delta$, and a number of circuit evaluations per inner product $R \geq \frac{4n^{2(t+c)}}{\delta}$, we have*

$$\Pr\left( \operatorname{argmin}_{x \in X} \widetilde{\mathcal{L}}(\log_g(x)) = \operatorname{argmin}_{x \in X} \mathcal{L}(\log_g(x)) \right) \geq 1 - \frac{\delta}{2}$$

*Proof.* In order for the minima of the two losses to be obtained for the same $x \in X$, it is sufficient to ensure that the classifiers $h_{\log_g(x_i)}$ and $\widetilde{h}_{\log_g(x_i)}$ agree on all points $x_j$, for all $(x_i, x_j) \in X$. This can be enforced by having:

$$\left( \bigcap_{\substack{i,j \\ i \neq j}} x_i \in I_{\log_g(x_j)} \right) \cap \left( \bigcap_{i,s'} |e_{i,s'}| \leq \frac{\Delta}{2} \right)$$

that is, having for all classifiers $h_{\log_g(x_j)}$ that all points $x_i \in X$, $x_i \neq x_j$, are away from its boundary regions in log-space, and that the labels assigned to these points are all the same under noise.
We bound the probability of the negation of this event:

$$\Pr\left( \bigcup_{\substack{i,j \\ i \neq j}} x_i \notin I_{\log_g(x_j)} \cup \bigcup_{i,s'} |e_{i,s'}| \geq \frac{\Delta}{2} \right) \leq \Pr\left( \bigcup_{\substack{i,j \\ i \neq j}} x_i \notin I_{\log_g(x_j)} \right) + \Pr\left( \bigcup_{i,s'} |e_{i,s'}| \geq \frac{\Delta}{2} \right)$$

using the union bound.
We start by bounding the first probability, again using the union bound:

$$\Pr\left( \bigcup_{\substack{i,j \\ i \neq j}} x_i \notin I_{\log_g(x_j)} \right) \leq \sum_{\substack{i,j \\ i \neq j}} \Pr\left( x_i \notin I_{\log_g(x_j)} \right)$$

$$= \sum_{\substack{i,j \\ i \neq j}} \frac{\Delta}{2} \leq \frac{n^{2c}\Delta}{2}$$

By setting $t \geq 3c$, we have $\Delta \leq 4n^{-t} \leq 4n^{-3c}$, which allows us to bound this first probability by $\delta/4$ when $c \geq \log_n(8/\delta)$.
As for the second probability above, we have

$$\Pr\left( \bigcup_{i,s'} |e_{i,s'}| \geq \frac{\Delta}{2} \right) \leq \sum_{i,s'} \Pr\left( |e_{i,s'}| \geq \frac{\Delta}{2} \right)$$

$$\leq \frac{4n^{2c}}{\Delta^2 R}$$

using the union bound and Eq. (20). By setting $R \geq \frac{4n^{2(t+c)}}{\delta} \geq \frac{16n^{2c}}{\Delta^2 \delta}$ (since $\Delta \geq 2n^{-t}$), we can bound this second probability by $\delta/4$ as well, which gives:

$$\Pr\left(\operatorname*{argmin}_{x \in X} \widetilde{\mathcal{L}}(\log_g(x)) = \operatorname*{argmin}_{x \in X} \mathcal{L}(\log_g(x))\right) \geq 1 - \Pr\left(\bigcup_{\substack{i,j \\ i \neq j}} x_i \notin I_{\log_g(x_j)} \cup \bigcup_{i,s'} |e_{i,s'}| \geq \frac{\Delta}{2}\right)$$

$$\geq 1 - \delta/2 \qquad \qquad \square$$

**Lemma 8.** *For a training set of size $n^c$ such that $c \geq \log_n\left(\frac{\log(\delta/2)}{\log(1-2\varepsilon)}\right)$, then $s' = \log_g\left(\operatorname{argmin}_{x \in X} \mathcal{L}(\log_g(x))\right)$ is within $\varepsilon$ distance of the optimal $s$ with probability:*

$$\Pr\left(\frac{|s'-s|}{p-1} \leq \varepsilon\right) \geq 1 - \frac{\delta}{2}$$

*Proof.* We achieve this result by proving:

$$\Pr\left(\frac{|s'-s|}{p-1} \geq \varepsilon\right) \leq \frac{\delta}{2}$$

This probability is precisely the probability that no $\log_g(x) \in \log_g(X)$ is within $\varepsilon$ distance of $s$, i.e.,

$$\Pr\left(\bigcap_{x \in X} \log(x) \notin [s - \varepsilon(p-1), s + \varepsilon(p-1)]\right)$$

As the elements of the training set are all i.i.d., we have that this probability is equal to

$$\Pr\left(\log(x) \notin [s - \varepsilon(p-1), s + \varepsilon(p-1)]\right)^{|X|}$$

Since all the datapoints are uniformly sampled from $\mathbb{Z}_p^*$, the probability that a datapoint is in any region of size $2\varepsilon(p-1)$ is just $2\varepsilon$. With the additional assumption that $|X| = n^c \geq \log_{1-2\varepsilon}(\delta/2)$ (and assuming $\varepsilon < 1/2$), we get:

$$\Pr\left(\frac{|s'-s|}{p-1} \geq \varepsilon\right) \leq (1 - 2\varepsilon)^{\log_{1-2\varepsilon}(\delta/2)} = \frac{\delta}{2} \qquad \qquad \square$$

Lemma 7 and Lemma 8 can be used to prove:

**Corollary 1.** *For a training set of size $n^c$ such that $c \geq \max\left\{\log_n(8/\delta), \log_n\left(\frac{\log(\delta/2)}{\log(1-2\varepsilon)}\right)\right\}$, a $t \geq 3c$ in the definition of $\Delta$, and a number of circuit evaluations per inner product $R \geq \frac{4n^{2(t+c)}}{\delta}$, then our optimization algorithm returns a variational parameter $g^{s'}$ such that*

$$\Pr\left(\frac{|s'-s|}{p-1} \leq \varepsilon\right) \geq 1 - \delta$$

From here, we notice that, when we apply Corollary 1 for $\varepsilon' \leq \frac{\Delta}{2}$, our optimization algorithm returns an $s'$ such that, with probability $1 - \delta$, the set $I_{s'}$ is equal to $I_s$ and is of size $(p-1)(1-2\Delta)$. In the event where $|s' - s|/(p-1) \leq \varepsilon' \leq \frac{\Delta}{2}$, we can hence bound the accuracy of the noisy classifier:

$$\operatorname{Acc}_{\widetilde{h}_{s'}}(f_s) = \frac{1}{p-1} \sum_{x \in \mathcal{X}} \Pr\left(\widetilde{h}_{s'}(x) = f_s(x)\right)$$

$$\geq \frac{1}{p-1} \sum_{x \in I_s} \Pr\left(\widetilde{h}_{s'}(x) = f_s(x)\right)$$

$$\geq (1 - 2\Delta) \min_{x_i \in I_s} \Pr\left(|e_{i,s'}| \leq \frac{\Delta}{2}\right)$$

$$\geq (1 - 2\Delta)\left(1 - \frac{4}{\Delta^2 R}\right)$$

$$= 1 - \left(2\Delta\left(1 - \frac{4}{\Delta^2 R}\right) + \frac{4}{\Delta^2 R}\right)$$

with probability $1 - \delta$.

We now set $t \geq \max\left\{3\log_n(8/\delta), \log_n(16/\varepsilon)\right\}$, $\varepsilon' = n^{-t}$ and $R \geq \max\left\{\frac{4n^{2(t+c)}}{\delta}, \frac{128}{\varepsilon^3}\right\}$, such that $2\varepsilon' = 2n^{-t} \leq \Delta \leq 4n^{-t} \leq \frac{\varepsilon}{4}$, $\left(1 - \frac{4}{\Delta^2 R}\right) \leq 1$ and $\frac{4}{\Delta^2 R} \leq \frac{\varepsilon}{2}$.

Using these inequalities, we get

$$\text{Acc}_{\widetilde{h}_{s'}}(f_s) \geq 1 - \varepsilon$$

with probability $1 - \delta$, which proves Theorem 2.