# OpenReview forum: "Parametrized Quantum Policies for Reinforcement Learning"
_NeurIPS.cc/2021/Conference — NeurIPS 2021 Poster_

### Official Review · Reviewer_e14U · 2021-07-13

**Rating:** 7
**Confidence:** 5

**Summary:**

## Summary

The paper proposes a policy-based training framework with variational quantum circuits (VQC). Although the VQC-based deep reinforcement learning algorithm has been first studied in [20] and [23], this paper further addresses a policy-based optimization instead of a value-based method (e.g., DQN) in [20].

The introduction and related work parts are good covering most of the important references in the field.

The paper further discusses two variants of the parameterized quantum circuit (PQC), raw-PQC and softmax-PQC to compared with a DNN-based RL method in three DRL environments from OpenAI gym.

- Noted I am not sure how fair is the selection DNN-based RL method in terms of model capacity according to the current version.

The authors further discuss some points included of (1) Efficient policy sampling and (2) quantum advantages.

Overall, the paper is well-written and motivated. The major weakness is on the baseline selection and some scalable discussion on quantum advantages.

- baseline selection

The works in [22] using a continuous quantum Deep RL method, called Quantum DDPG. What is the advantage between the proposed softmax PQC compared with Quantum DDPG?

- scalable discussion

I just notice that the value-based method paper [20] uses a simple tangible radio RL environment to discuss the trade-off between parameter complexity (Figure 11 in [20]) versus different classical and quantum RL methods.

If the author could provide experimental supports or discussion versus that existing analysis on the QA on parameters, the discussion could be more comprehensive.




**Limitations And Societal Impact:**

- The checklist is not fully satisfied on 1. (c) Did you discuss any potential negative social impacts of your work? [No]

The authors may consider adding a board impact section in their revised supplementary.

**Main Review:**

## Pros

- The paper provides a policy-based method for quantum deep reinforcement learning.

- The discussion and related works are of good quality.

- The results show the capability of Quantum DRL working on new challenges environment as Mountains Car (initial with negative values) and Acrobot-v1.

## Points to Improve

- The quantum advantage needs some empirical analysis [Ref 1] or learning complexity analysis (PAC theory) to have a full claim.
Please refer to the potential scalable discussion in the summary to have parameter complexity improvement.

- The DNN policies in Figure 5 are associated with the same amount of trainable parameters or? This would need more clarification. List or provide a parameter table for each baseline would be a solution.


## Addtional Disccusion

- There is less discussion on the full checklist, such as potential energy consumption (CPU-simulated or using QPU batch directly) corresponding to the NeurIPS policy.

- If the proposed method could also benefit from a decentralized framework as the recent studies [Ref 2] in a hybrid classical-quantum A2C policy?


Overall, the paper is around the borderline acceptance threshold.

I give my recommendation to borderline acceptance considering its pros, weaknesses, and detailed supplementary report. I would consider modifying my score based on the additional baseline studies and discussion in a revised version.




Ref 1. A Grover-search based quantum learning scheme for classification, New Journal of Physics, 2021

Ref 2. Decentralizing feature extraction with quantum convolutional neural network for automatic speech recognition, IEEE ICASSP 2021

Ref 3. Quantum-enhanced machine learning, Physical review letters 2016

**Time Spent Reviewing:**

20

---

> ### Author Response · Authors · 2021-08-09
> **Authors response**
>
> We thank the reviewer for their careful assessment of our manuscript and for their particular attention to ways in which we could improve our work. The reviewer’s suggestions mainly concern our baseline discussion and our quantum advantage results. As we detail point by point below, we believe that most of these suggestions have already been addressed in various parts of our manuscript:
>
> - Regarding the learning complexity analysis of our quantum advantage environments, we have already provided a complete theoretical analysis in Appendix F to J, which proves a clear learning advantage of our PQC learners over any efficient classical algorithm.
>
> - For the comparison of learning performance of our PQCs and standard DNNs, we attempt to make our analysis fair with a thorough hyperparametrization of the DNNs and providing them with an unrestricted number of parameters. All hyperparameters are listed in Appendix C.
>
> - With respect to a comparison with other quantum models for RL, we note that previous works using PQCs have been unsuccessful in solving the benchmarking environments we consider. The Ref. [20] that the reviewer mentions specifically studies a different quantum learning setting, that does not allow us to compare their models to ours.
>
> If we did not fully appreciate the concerns of the reviewer we would be happy to readdress them in the discussion round.
>
> We do note that most of the questions raised by the reviewer are addressed only in the appendix, and we are open to moving some of those results to the main text.
>
> More detailed responses are given below:
>
> **Re: Learning complexity analysis for quantum advantage**
>
> > The quantum advantage needs some empirical analysis [Ref 1] or learning complexity analysis (PAC theory) to have a full claim.
>
> The reviewer suggests that we should provide a learning complexity analysis for the DLP environments that we consider, and [Ref. 1] is given as a potential approach to prove a separation in query complexity. However, this point has already been addressed in Appendix J (more specifically Theorem 2), where we prove the existence of a learning algorithm using a polynomially-large training set and polynomial-time processing to find a close-to-optimal policy in any of our DLP environments with high probability. This result involves a PAC-type analysis, and also takes into account noise considerations in the evaluation of PQC expectation values. Together with the classical hardness results of Appendix F to H, this also constitutes a proof of exponential separation (or quantum speed-up) in query and time complexity with any efficient classical algorithm attempting to solve these learning problems.
>
> **Re: Parameter complexity**
>
> > I just notice that the value-based method paper [20] uses a simple tangible radio RL environment to discuss the trade-off between parameter complexity (Figure 11 in [20]) versus different classical and quantum RL methods.
>
> The reviewer also suggests that we should conduct a parameter complexity comparison between our PQCs and, e.g., DNNs in solving the several environments we consider in our numerical analysis. However, from the hyperparameter list we provide in Appendix C (Table 3), one can see that the number of parameters used by our PQCs is relatively low (less than 100 to achieve close-to-optimal performance in all environments using softmax-PQCs), which is notably much lower than the best performing DNN we found in the SL-PQC and Cliffwalk-PQC environments (902, where the weights between hidden layers have a large contribution). Given the small size of these environments, we were not sure how to interpret these results, which explains why we did not include them in the discussion. As for studying a parameter scaling behavior, this would require us to consider different environments, which is beyond the scope of this paper.
>
> > The DNN policies in Figure 5 are associated with the same amount of trainable parameters or? This would need more clarification. List or provide a parameter table for each baseline would be a solution
>
> The final number of trainable parameters used in these experiments are listed in this same Table 3 in Appendix C, along with the other hyperparameters of this baseline. As for the DNN architectures tested in our hyperparameter search, these are also specified in Section 4.2.2. (a grid search over depths 2-10 and widths 8-64, which corresponds to a number of parameters between 114 and 37762).
>
> > Noted I am not sure how fair is the selection DNN-based RL method in terms of model capacity according to the current version.
>
> Taking into account the number of parameters considered in our hyperparametrization (see above), the DNNs have been selected with an unrestricted model capacity.
>
> We also note that a parameter complexity discussion has also been conducted in our concurrent manuscript, Quantum agents in the Gym (included in the Supplementary Material), where PQCs similar to ours, albeit trained in a value-based manner, show an advantage in number of parameters compared to standard DNNs in the CartPole environment.
>
> **Re: Comparison to quantum DDPG**
>
> > The works in [22] using a continuous quantum Deep RL method, called Quantum DDPG. What is the advantage between the proposed softmax PQC compared with Quantum DDPG?
>
> The Quantum DDPG model takes as input genuine quantum states (including superpositions) and produces continuous actions. It has therefore a completely different domain of applicability than the softmax-PQC, which takes as input (continuous) classical states and only works in environments with discrete action spaces. We cannot therefore compare these models on the same tasks. As far as the action space is concerned however, one can also adapt our PQC design to output a continuous action, expressed as an expectation value at its output (same as for the DDPG), and train this policy using the classical DDPG algorithm.
>
>
> We also address three additional points mentioned by the reviewer:
>
> **Re: Energy consumption**
>
> > There is less discussion on the full checklist, such as potential energy consumption (CPU-simulated or using QPU batch directly) corresponding to the NeurIPS policy
>
> We do list the computing resources that we used to run our simulations in Appendix C (estimated 20 000 total CPU-hours). Note however that this estimate covers the compute time of the full hyperparametrization we performed and that the TFQ code accompanying our work gives a final runtime of ~40min CPU-time to train a single PQC agent on CartPole. From this, one could derive the energy consumption of this training and compare it to that of classical algorithms. Further investigation of energy consumption on QPUs constitutes an interesting question for future research.
>
> **Re: Decentralized framework**
>
> > If the proposed method could also benefit from a decentralized framework as the recent studies [Ref 2] in a hybrid classical-quantum A2C policy?
>
> This is an interesting avenue for further investigation, and we do not see a reason why it could not be considered. But this seems beyond the scope of our paper, which focuses on a simple policy-based approach for which we show that it can achieve a similar performance to DNNs in benchmarking environments, and further prove a quantum learning advantage in constructed environments.
>
> **Re: Board impact section**
>
> > The authors may consider adding a board impact section in their revised supplementary.
>
> When writing the manuscript, it was not clear to us what we could mention in such a broad impact section, but we would be glad to add one where we would discuss the implications of our learning advantage results and prospects of PQCs in RL.
>
> The reviewer graciously ends their review with: :
>
> > I would consider modifying my score based on the additional baseline studies and discussion in a revised version.
>
> In the above, we have provided a full account of how we feel we have provided some if not most of the analyses the reviewer suggested. Furthermore, we are also happy to include parts of the discussion above in the final manuscript. We thus hope this discussion will inspire the reviewer to favorably amend the rating of our paper.

---

> > ### Comment · Reviewer_e14U · 2021-08-09
> > **Re: Authors response to the Reviewer e14U**
> >
> > Thank the authors to provide additional discussion and insights. Here are some follow-up discussions after reading the authors' responses. Noted I had read the appendix of the submission 8446 in the first round review.
> >
> > By additional connections provided by the authors, some of my concerns have been resolved.
> >
> > This follow-up discussion aims to provide some additional information and space that would benefit the paper in a final revised format.
> >
> > I would encourage the authors to be open to improve some of the related content. It is a little bit defensive by reading the current authors' feedbacks.
> >
> > ---
> >
> > **Re: Ans. 1. When writing the manuscript, it was not clear to us what we could mention in such a "broad impact section."**
> >
> > I could recommend reading the official broad impact guideline in the previous NeurIPS FAQ [Ref A]. It would be better to include a related section with a few lines to summarize the related "broad impacts" (e.g., social) in the main paper or appendix in a revised version. This would enlarge the potential audience for this work considering its nature to be interdisciplinary.
> >
> > e.g., `How should I write the Broader Impact section? Answer: For additional motivation and general guidance, read Brent Hecht et al.’s white paper and blogpost, as well as this blogpost from the Centre for Governance of AI. For an example of such a discussion, see sec. 4 in this paper from Gillick et al.`
> >
> > Also noted giving negative checklist feedback is not ideal. I suggest the authors may reconsider including a related section to enlarge the general impacts of this work.
> >
> > ---
> >
> > **Re: Ans. 2. "We do note that most of the questions raised by the reviewer are addressed only in the appendix, and we are open to moving some of those results to the main text."**
> >
> > Yes, that would be a better modification. Please also provide the specific hyper-reference indexes and a short summary of the appendix findings. The connections are not clear and the related writing style would be very improved.
> >
> > ---
> >
> > **Re: Ans. 3. Related to [20] and "As for studying a parameter scaling behavior, this would require us to consider different environments, which is beyond the scope of this paper."**
> >
> > I was expecting to see a simple reproducible study of the proposed PQC framework on that communication radio experiment in [20] (e.g., just 2 to 4 observations variables).
> >
> > Considering most of the currently evaluated environments are most game-based simulators, including such a real-world-oriented application would increase the impact and empower sim2real settings for the general NeurIPS audience.
> >
> > ---
> >
> > Thanks for the additional discussion. Please consider to include the added discussion in a future revised version.
> >
> > Ref A. NeurIPS 2020 FAQ for Authors, Oct 2020
> > https://nips.cc/Conferences/2020/PaperInformation/NeurIPS-FAQ

---

> > > ### Author Response · Authors · 2021-08-27
> > > **Re: Re: Authors response to the Reviewer e14U**
> > >
> > > We thank the reviewer for further clarifying some of their concerns. We are happy to follow up the discussion below and to make our suggested modifications more concrete.
> > >
> > > ---
> > >
> > > **Re: Ans. 1**
> > >
> > > We thank the reviewer for pointing us to the guidelines and general guidance in writing a broad impact section. We agree that such a section would benefit the paper and we have therefore drafted the following paragraphs, that we would be glad to add to the revised manuscript.
> > > ```
> > > We expect our work to have an overall positive societal impact. Notably, we believe that our approach to QRL could be beneficial in the two following ways:
> > > -	Modern-day RL is known to be very resource-heavy in terms of compute power and energy consumption (see, e.g., the resources needed to train AlphaGo Zero [1]). In other computational problems, e.g., the quantum supremacy problem of Google [2], it was shown that, because of their computational advantages, quantum computers could save many orders of magnitude in energy consumption compared to classical supercomputers [3]. Therefore, a quantum learning advantage as showcased in our work could potentially alleviate the computational demands of RL, making it more economically appealing and environmentally-friendly.
> > > -	Aside from the game-based problems that we consider in our work, the areas of application of RL are constantly increasing [4-6]. The learning advantages of QRL could potentially make these existing applications more accessible technologically and economically, but also unlock new applications, e.g., in problems in quantum information [7, 8] or quantum chemistry [9].
> > > At the same time, this work may have some negative consequences. Notably, QRL will inherit many of the problems that are already present in classical RL and ML in general. For instance, it is not clear whether the question of interpretability of learning models [10] will be negatively or positively impacted by switching to quantum models. One could argue that the inability to fully access the quantum Hilbert spaces in which quantum computers operate can turn learning models even further into “black-boxes” than existing classical models. Also, similarly to the fact that current state-of-the-art ML/RL requires supercomputers that are not accessible to everybody, privatized and select access to quantum computers could emphasize existing inequalities in developing and utilizing AI.
> > >
> > > [1] Huang, Dan, “How much did AlphaGo Zero cost?” Blogpost: https://www.yuzeh.com/data/agz-cost.html
> > > [2] Arute, Frank, et al. "Quantum supremacy using a programmable superconducting processor." Nature 574.7779 (2019): 505-510.
> > > [3] Villalonga, Benjamin, et al. "Establishing the quantum supremacy frontier with a 281 Pflop/s simulation." Quantum Science and Technology 5.3 (2020): 034003.
> > > [4] Kober, Jens, J. Andrew Bagnell, and Jan Peters. "Reinforcement learning in robotics: A survey." The International Journal of Robotics Research 32.11 (2013): 1238-1274.
> > > [5] Mahmud, Mufti, et al. "Applications of deep learning and reinforcement learning to biological data." IEEE transactions on neural networks and learning systems 29.6 (2018): 2063-2079.
> > > [6] Yu, Chao, Jiming Liu, and Shamim Nemati. "Reinforcement learning in healthcare: A survey." arXiv preprint arXiv:1908.08796 (2019).
> > > [7] Albarrán-Arriagada, Francisco, et al. "Measurement-based adaptation protocol with quantum reinforcement learning." Physical Review A 98.4 (2018): 042315.
> > > [8] Wu, Shaojun, et al. "Quantum reinforcement learning in continuous action space." arXiv preprint arXiv:2012.10711 (2020).
> > > [9] Peruzzo, Alberto, et al. "A variational eigenvalue solver on a photonic quantum processor." Nature communications 5.1 (2014): 1-7.
> > > [10] Linardatos, Pantelis, Vasilis Papastefanopoulos, and Sotiris Kotsiantis. "Explainable ai: A review of machine learning interpretability methods." Entropy 23.1 (2021): 18.
> > > ```
> > >
> > > ---
> > >
> > > **Re: Ans. 2**
> > >
> > > >> "we are open to moving some of those results to the main text."
> > >
> > > > Yes, that would be a better modification.
> > >
> > > Given that our result on learning complexity analysis appears as the most susceptible to be overlooked, we are happy to add an informal statement of Theorem 2 (Appendix J) to the main text (Sec. 4.1.).
> > >
> > > > Please also provide the specific hyper-reference indexes and a short summary of the appendix findings.
> > >
> > > We are also happy to provide such hyper-references throughout the main text, and add a short summary of the results in the appendix as to make them more accessible.
> > > Moving more results to the main text would unfortunately cause us to exceed the page limit of NeurIPS submissions. We hope nonetheless that these proposed modifications agree with what the reviewer has in mind.
> > >
> > > ---
> > >
> > > **Re: Ans. 3**
> > >
> > > > Considering most of the currently evaluated environments are most game-based simulators, including such a real-world-oriented application would increase the impact and empower sim2real settings for the general NeurIPS audience.
> > >
> > > As to enrich our numerical study with a real-world-oriented application, we would gladly show that our approach is also capable of handling the communication radio experiment in [20], for different instance sizes. In the time since our previous response, we have successfully trained our PQCs on this environment (with 2 to 5 observations variables) using a scaling of number of parameters that matches that of the PQCs in [20], i.e., O(n). We will include this numerical study, which also connects better to previous literature, to an appendix of the revised manuscript.

---

> > > > ### Comment · Reviewer_e14U · 2021-08-30
> > > > **Re: Final Recommendation**
> > > >
> > > > I have read the author's responses and their follow-up discussion with the other reviewers carefully.
> > > >
> > > > Considering that major algorithm foundations and superior results do not have major changes, I tend to accept this paper owing to its novel policy-based QVC format and foundations.
> > > >
> > > > In general, if the authors will **follow their words** and make all the related minor modifications (e.g., border impact section and connection to some real-world Q-learning applications) in a revised version, I tend to increase my score to 7 considering its novelty and quality on advancing QVC designs to hard Q-learning problems.
> > > >
> > > > This would be my final recommendation. Thanks.

---

### Official Review · Reviewer_fBf5 · 2021-07-16

**Rating:** 6
**Confidence:** 5

**Summary:**

The article titled ‘Parametrized Quantum Policies for Reinforcement Learning’ presents a hybrid quantum-classical learning model, which the author claims that can be effectively trained to solve several standard benchmarking environments. The learning model is based on parametrized quantum circuits (PQC), where the parameters can be adjusted during the training process. By defining RAW- and SOFTMAX-PQC, the author shows the detailed learning methods corresponding each of the cases. The work also displays the result of this two methods, which we can see a moderate learning process is happening. In the end, the author also argues the ability of parametrized quantum circuits to solve certain learning tasks that are intractable to classical models.


**Limitations And Societal Impact:**

As for the limitations and societal impact, the author argues that there exist RL environments where PQC policies can provide a learning advantage over standard classical policies. However, it looks like the argument is mostly change of narratives based on previous results and the contribution is rare.

**Main Review:**

The article is written nicely and the it is easy to follow the construction. However, I think there are two major drawbacks in this work. Firstly, in the RAW-PQC design, the policy is output of the quantum circuit. The probabilities of different strings are first evaluated and the an action is sampled from this distribution. This is redundant in my opinion because a quantum circuit is already a good sampler. Even though the author proves it is efficient to sample in this case, one can just use the measurement result each time as the sample of the distribution. So I do not see an obvious reason why the author need to evaluate the probability distribution first and then do the sampling. Secondly, in the SOFTMAX-PQC design, the policy is determined by the weighted mean value in line 112. It seems nice but there is a fatal drawback here. With this design, it may happen that the classical optimizer can optimize the weight \omega to get whatever probability distribution, even with the quantum circuit always producing the same output at all times. In this sense, the information that the quantum circuit is supposed to convey is absorbed into the classical parameters \omega.


**Time Spent Reviewing:**

24

---

> ### Author Response · Authors · 2021-08-09
> **Authors response**
>
> We thank the reviewer for raising their legitimate concerns regarding the models we define. However, we respectfully disagree that any of them constitute a "fatal drawback" as the reviewer suggests, but are more likely a matter of confusion perhaps due to our suboptimal explanation of the results and ideas.
>
> **Re: Use of raw-PQC policies**
>
> The reviewer seems to be confused as to when the output distribution of the raw-PQC is to be estimated. They claim that we rely on such an estimation to sample actions from the PQC:
>
> > The probabilities of different strings are first evaluated and the an action is sampled from this distribution. This is redundant in my opinion because a quantum circuit is already a good sampler.
>
> However, this is not the method that we suggest for the mere **sampling** from the raw-PQC policy. Indeed we suggest to sample actions using single measurements, exactly as the reviewer proposes.
>
> However with respect to this point:
>
> > So I do not see an obvious reason why the author need to evaluate the probability distribution first and then do the sampling.
>
> unfortunately, there is a strong reason. The estimating of the PQC output distribution, i.e., the expectation values $\langle P_a \rangle_{s,\theta}$, turns out to be necessary **during training** of the raw-PQC policy, as computing the log-policy gradient requires these estimates (see Eq. (6), line 132).
>
> However, we would like to stress that, unlike in many applications where the estimation of a distribution (to some error in $\ell_1$-norm) may nullify any quantum advantage, it is not the case here because this distribution is over $|A|$ (i.e., number of actions) elements (the bitstrings associated to a given action are considered undistinguishable), which allows it to be estimated in time at most linear in $|A|$, even to small error in $\ell_1$-norm (see [Ref 1], Table 1).
>
> **Re: "Quantumness" of softmax-PQC**
>
> The second "fatal drawback" that the reviewer puts forward is in the potential lack of "quantumness", or the importance of the classical parts in the softmax-PQC model, that could, ostensibly, render the quantum circuit pointless. This concern arises when looking into the post-processing of the output expectation values of the softmax-PQC model, which are weighted by trainable parameters (\omega):
>
> > in the SOFTMAX-PQC design, the policy is determined by the weighted mean value in line 112.
>
> With this design, the reviewer claims that it could happen that
>
> > the quantum circuit always producing the same output at all times
>
> would still allow the softmax-PQC policy to be trained arbitrarily, since
>
> > the classical optimizer can optimize the weight \omega to get whatever probability distribution.
>
> We likely carry some of the responsibility for this misreading of the model, and will happily do a better job at clarifying the situation in the revised version of the paper.
>
> The issue is, this reasoning is missing on the important point that the weights \omega are independent of the input data $s$, which, in the case where the PQC would produce a fixed output (i.e., independent of $s$), only trivial (state-independent) policies would be obtained. The fact that the data encoding only happens in the encoding gates of the PQC (and hence is entirely done in quantum feature space) hence means that the PQC will always have a non-trivial contribution to the model and the weights \omega are only here to enhance its expressivity.
>
> To give a better representation of what these weights are doing to the output of the PQC, we clarify here their specific action, which we would be glad to explain better in a revised version of the manuscript. Note first, that the operators $H_{a,i}$ in line 112 that we use in our numerical simulations are all mutually commuting and diagonal in the diagonal basis (see Table 3 in the Appendix). Hence, all that the weights \omega do is weight computational basis states (in an under-parametrized way), as to adjust their global contribution to the expectation values. Again, these weights are data-independent and play a complementary role to the data-encoding and variational parts of the PQC (which prepare the output distribution of the PQC in a data-dependent way). As long as the unweighted expectation values $\langle H_{a,i}\rangle$ are hard to estimate classically, the PQC is hence a non-trivial quantum model. This same reasoning applies to quantum kernel models (also known as implicit quantum classifiers [7]), where quantum kernel evaluations (which are data-dependent expectation values) are weighted entirely classically as to produce a classifier. But for quantum kernels that are hard to estimate classically, these remain genuine quantum models.
>
> **Re: Contributions in quantum advantage results**
>
> One last worry of the reviewer had to do with the importance of our contributions in the quantum advantage results. More specifically, the reviewer claims that:
>
> > it looks like the argument is mostly change of narratives based on previous results and the contribution is rare.
>
> We respectfully disagree that our results are just a change of narratives as compared to previous results. In this work, we introduce an application of a modified type PQCs (and the modification was vital!) in a new context, with new both theoretical and numerical contributions in proving their learning advantages. We list some of the highlights below for the convenience of the reviewer:
>
> - A rigorous proof of a learning advantage over any efficient classical learner in genuine MDPs, based on the DLP (Appendix F to H),
>
> - A learning algorithm with rigorous performance guarantees to train PQCs in these environments (Appendix I and J),
>
> - Numerical evidence of a learning advantage over DNNs in PQC-generated environments (Section 4.2.2 and Appendix D).
>
> We hope this summary of our contributions inspires the reviewer to rate our work a bit more favorably.
>
> [Ref 1] Joran van Apeldoorn, Quantum Probability Oracles & Multidimensional Amplitude Estimation, TQC 2021, https://drops.dagstuhl.de/opus/volltexte/2021/14004/

---

> > ### Comment · Reviewer_fBf5 · 2021-08-27
> > **Partially agree but not all**
> >
> > Thanks the authors for clarifying the questions I raised in previous comments.
> >
> > For the problem in the raw-PQC policy, the author has explained that they sample from single measurement of the PQC, I accept this but I suggest the author to stress it more explicitly in the main text.
> >
> > Next, for the softmax-PQC policy, I don’t think the authors have give a convincing explanation. Even though the PQC may not be easy to simulate classically, it does not dismiss the possibility that the parameters \omega could fabricate an arbitrary probability distribution with an even classical output from the PQC. I suggest the author to elaborated more on how the PQC is indispensable if the parameters \omega are there.
> >
> > Thanks!

---

> > > ### Author Response · Authors · 2021-08-27
> > > **Further clarifications on softmax-PQCs**
> > >
> > > We thank the reviewer for the continued interesting discussion, and we are glad that they accept our explanation of the sampling procedure in the raw-PQC model. We agree that this is an important point, and we will gladly stress it in the revised manuscript.
> > >
> > > We also thank the reviewer for further clarifying their concern regarding the softmax-PQC model. We are happy to provide further justifications for our claims.
> > >
> > > The reviewer clarifies that the problem that concerns them most is not the "quantumness" of the model in general, but specifically
> > >
> > > > the possibility that the parameters \omega could fabricate an arbitrary probability distribution with an even classical output from the PQC.
> > >
> > > We understand now that the original question of the referee may be in fact quite broad. In order to give a precise response, we feel the need to clarify our interpretations of the question before giving our answer for each reading. We identify two interpretations of the reviewer’s question:
> > > 1.	The reviewer is asking if, in our particular use of the softmax-PQC model, where the operators $H_{a,i}$ appearing in line 112 are all diagonal in the computational basis, it could happen that all learning in the model would occur in the parameters $\omega_{a,i}$ weighting these operators, while the variational parts of the PQC would remain (arbitrarily) fixed.
> > > 2.	The reviewer is asking if, in the most general definition of our softmax-PQC model, where the weighted operators $\sum_{i} \omega_{a,i} H_{a,i}$ are rich enough to represent **any** observable, this same phenomenon could happen.
> > >
> > > We answer both interpretations.
> > >
> > > ---
> > >
> > > **Case 1**
> > >
> > > In this case, the phenomenon that the reviewer suggests cannot happen because only tweaking the observables weights is simply not expressive enough. We illustrate this below.
> > >
> > > Recall that the softmax-PQC is modelling a conditional probability distribution over actions given states $\pi(a|s)$. Let us first consider an example of a softmax-PQC composed of:
> > > -	an arbitrary PQC that encodes and variationally processes the states $s$,
> > > -	single operators $H_a = P_a$ for each action $a$, where $P_a$ is a projector on a single computational basis state associated to $a$, with $P_a P_{a’} = \delta_{a,a’}$,
> > > -	weights $\omega_a$ for each $H_a = P_a$.
> > >
> > > The softmax-PQC is therefore $\pi(a|s)=\frac{e^{\omega_a \langle P_a \rangle_{s,\theta}}} {\sum_{a’} e^{\omega_{a’} \langle P_{a’} \rangle_{s,\theta}}}$.
> > >
> > > Consider now that we fix $\theta$, leaving us with some fixed $\langle P_a \rangle_{s,\theta}$ for each $s$. We want to train solely the weights $\omega$ as to obtain an arbitrary $\pi(a|s)$. Clearly, because the $\omega$’s are independent of $s$, the only degrees of freedom we are left with are in changing the global magnitude of each $\langle P_a \rangle_{s,\theta}$ **independently of** $s$. While, for a fixed $s$, this would suffice to get an arbitrary $(\pi(a_1|s), \ldots, \pi(a_M|s))$ for that $s$ (assuming all $\langle P_a \rangle_{s,\theta}$ are non-zero), there are clearly not enough degrees of freedom to obtain an **arbitrary** policy $\pi(a|s)$ for all $s,a$ by solely controlling $\omega$. Therefore this makes the PQC indispensable.
> > >
> > > Now note that the operators $H_{a,i}$ that we use are in some cases more general than projectors on computational basis states. But, as we clarified in our previous response, we always restrict ourselves to operators that are diagonal in the computational basis, which means they can be decomposed as linear combinations of such projectors (e.g., a Pauli $Z = P_0 - P_1$, and tensor products with other Pauli $Z$ or $I$ give a sum of projectors on all computational basis states). Our parameters $\omega$ also weight computational basis states in an under-parametrized way, meaning one $\omega_{a,i}$ can effectively weight more than one computational basis state per action, and a given computational basis state can be weighted by more than one $\omega_{a,i}$ per action. For simplicity, let us consider the most expressive case where all and each computational basis state are weighted by a separate $\omega_{a,i}$ per action. Now, even in this case, the same observation remains that the weights $\omega$ do not accommodate enough degrees of freedom to represent arbitrary conditional probability distributions $\pi(a|s)$ **for each** $s$. This is simply because $\omega$’s are independent of $s$ and we consider continuous state space $S \ni s$ that do not allow $\omega$’s to capture all probability distributions $\pi(a|s)$ for all $s \in S$.
> > >
> > > ---
> > >
> > > **Case 2**
> > >
> > > The second case is much broader and more complicated, that is why we will not go fully into details here, but we can mention the following:
> > >
> > > In the case where one allows the weighted operators $\sum_{i} \omega_{a,i} H_{a,i}$ to parametrize **arbitrary** observables (i.e., using a number of terms that is exponential in the number of qubits) and where one allows very rich data encodings in the PQC (using, e.g., an encoding that provides a full explicit binary representation of the input vector $s$ using very many qubits), the phenomenon that the reviewer suggests could indeed happen.
> > > This is because, in this specific case, the observable could even encode an entire quantum computation on its own and therefore be universal (this is not the case if, e.g., we only allow commuting terms in the specification of the observable). But this would constitute an extremely expressive model, with an extremely large number of parameters, and it is unclear how valuable it would be in a ML/RL context.
> > >
> > > Many interesting cases could exist between the extreme scenario discussed here and that of Case 1 (i.e., in our work), where various restrictions could either allow or not the phenomenon mentioned by the reviewer, but we believe that this study is beyond the scope of this paper. We are currently preparing a follow-up paper that touches this specific subject.
> > >
> > > > I suggest the author to elaborated more on how the PQC is indispensable if the parameters \omega are there.
> > >
> > > We hope that this constitutes sufficient justification for the reviewer.

---

### Official Review · Reviewer_3zS5 · 2021-07-21

**Rating:** 6
**Confidence:** 2

**Summary:**

I thank the authors for their response. I think the empirical results are encouraging and the idea is reasonably novel, so will keep the score as is.

---
The paper discusses a parametrized quantum computations as a policy trained via policy gradient. The policy is parametrized with a quantum computer whereas parameter updates are done by a classical computer. Situations where the quantum policy can have advantage over classical ones are discussed and evaluated. In PQC-generated environments, PQC-policies have a significant advantage over classical ones.

**Ethical Concerns:**

No.

**Limitations And Societal Impact:**

Yes.

No negative societal impact is discussed, but I don't think it is necessary for this paper.

**Main Review:**

Originality: The approach taken is generally a standard one taken with quantum machine learning -- a quantum circuit is parametrized and optimized via a classical update algorithm. Since gradients of the quantum circuit can be evaluated using e.g. the parameter shift trick, it is not entirely surprising for a policy gradient algorithm to be developed.

The paper mentioned some recent related work on this front, but it would also be interesting to discuss older related works (https://arxiv.org/abs/0810.3828, https://arxiv.org/abs/1401.4997, https://arxiv.org/abs/1612.05695). One contribution is the introduction of softmax-PQC which clearly outperforms raw-PQC policies.

Applicability: The PQC-constructed environment is largely an MDP-extension of Liu et al., and it does not seem that the action in the PQC-cliffwalk affects the trajectory other than episode termination. The PQC policy is limited to discrete actions and a comparison with classical algorithms is not included.

Clarity: the paper is written very well and is about readable as it can get for an audience that knows ML but not quantum computing.

Minor questions and or comments:
- How does the TensorFlow quantum environment simulate quantum noise (or does it)? I would suppose a larger noise would hinder learning progress even though policy gradient should be able to learn with more episodes.
- It seems to me that the major advantage of PQC is the ability to learn a map for DLP -- are there other more realistic examples where PQC policies are needed?
- It seems that the softmax-PQC is not limited to reinforcement learning. Do you expect it to achieve similar improvements in supervised learning?

**Time Spent Reviewing:**

3h

---

> ### Author Response · Authors · 2021-08-09
> **Auhtors response**
>
> We thank the reviewer for their thoughtful review of our manuscript and for their relevant questions regarding the prospects of our work. Before answering these questions, we would like to clarify an issue regarding our quantum advantage results, which may have caused confusion with respect to the applicability of our methods.
>
> **Quantum advantage in RL**
>
> In our paper, we in fact study two classes of quantum advantage in RL: one is rigorously proven and can be seen mainly as MDP-extensions of the DLP task of Liu et al. The other class of separations that we investigate are more pragmatic, and are supported by numerical evidence only. These rely on our (raw-) PQCs to specify the RL environments (rather than basing them on DLP).
>
> In light of this, we would like to address the following comment of the reviewer, which left us suspecting a possible misinterpretation.
>
> > The PQC-constructed environment is largely an MDP-extension of Liu et al.
>
> The DLP-environments mentioned above are indeed MDP-extensions of Liu et al. But the PQC-generated environments are not. Furthermore we highlight that, apart from the SL-DLP environment where the quantum advantage results of Liu et al. are trivially derived, our more interesting MDP extensions (involving temporal structure and determinism) have non-trivial proofs of separation, as we detail in Appendix F to H. Finally, just for clarity, our overall methods are of course meant to be used beyond the scope of these very special environments that we study only for the purpose of providing (varying levels) of evidence of the potential for quantum advantages
>
> > a comparison with classical algorithms is not included.
>
> For the case of PQC-generated environments we do provide a numerical comparison between our PQCs and standard DNNs. With respect to our results in the DLP environments, we do not do a numerical investigation since we formally prove an (asymptotic) separation between our PQC agents and any polynomial-time classical algorithm (assuming the widely-believed hardness of DLP).
>
> > it does not seem that the action in the PQC-cliffwalk affects the trajectory other than episode termination. The PQC policy is limited to discrete actions
>
> Our PQC-cliffwalk environment is indeed of little practical interest on its own and does not present any special structural complexity. It is, however, sufficiently complex to showcase an RL environment where our PQC policies show an empirical advantage over standard DNNs used in deep RL. As mentioned, this does not limit the applicability of our approach, and our PQC policies can be used in any classical RL environments with continuous/large state spaces and discrete action spaces.
>
> We now move to the direct questions of the reviewer:
>
> **Re: Noise in simulations**
>
> > How does the TensorFlow quantum environment simulate quantum noise (or does it)?
>
> The TensorFlow Quantum simulations that we use do not incorporate noise in the analyses that we have performed. This is indeed a very interesting research avenue, that would absolutely strengthen our claim of compatibility with NISQ devices. However, in the field of QML, most papers address only one of the three main issues: getting a method to work and characterizing it in broad strokes (with or without numerics) [5], providing theory regarding quantum/classical separations [14] or analyzing robustness to noise (usually happens only with more established methods). Very few [6] do more than one. This is why we felt introducing a new method, providing analysis of practical performance, and proving a separation suffice for one of the first papers on QRL with PQCs, whereas any analysis of noise robustness (which is a multi-parameter and very involved issue) would overload the focus or our message. We however agree with the reviewer that noise would probably hinder (but given limitations, not totally forbid) learning.
>
> **Re: Applicability**
>
> > It seems to me that the major advantage of PQC is the ability to learn a map for DLP -- are there other more realistic examples where PQC policies are needed?
>
> As explained above, the DLP environments were only of interest to us from a theoretical perspective to demonstrate the potential of PQC policies for RL. Our PQC-generated environments already illustrate more pragmatic environments that are more natural for our PQC policies than standard DNNs. Given the supporting evidence that PQC functions are fundamentally different in nature from DNNs [33], we have reasonable expectations for them to be practically advantageous in different domains as well. Such domains would be, for example, those where the PQC design can be made more problem-specific. This could happen, e.g., in quantum chemistry, where variational quantum eigensolvers can use the UCCSD Ansatz, or some quantum alternating operator Ansätze.
>
> **Re: softmax-PQC outside RL**
>
> > It seems that the softmax-PQC is not limited to reinforcement learning. Do you expect it to achieve similar improvements in supervised learning?
>
> Despite its name, the softmax-PQC model introduces 2 novel aspects: 1) the softmax non-linearity; but also 2) trainable observables.
>
> We comment on their relevance in opposite order:
>
> 2) We have compelling evidence that this aspect helps get more expressive and faster-learning PQCs in regression tasks. Indeed, we observe an associated boost in performance in our related work, Quantum agents in the Gym (in the Supplementary Material), where the value-based approach we take for PQC-based RL is arguably a (hard) regression task.
>
> 1. We believe that the softmax non-linearity won’t help much in classification tasks where threshold functions (i.e., argmax) trump softmax, but do agree that it can be helpful in unsupervised learning tasks such as generative modeling, where a tunable “peakness" of distribution can be advantageous.
>
> Finally, we address two other remarks of the reviewer:
>
> **Re: Related work**
>
> > it would also be interesting to discuss older related works (https://arxiv.org/abs/0810.3828, https://arxiv.org/abs/1401.4997, https://arxiv.org/abs/1612.05695)
>
> These works discuss different types of models for quantum-enhanced RL (Grover-based and QBMs), but they are related so we would be happy to include them in the discussion along Ref. [23].
>
> **Re: Originality**
>
> As the reviewer suggests,
>
> > it is not entirely surprising for a policy gradient algorithm to be developed.
>
> and the idea of using PQCs for RL was indeed put forward relatively early on in the history of PQCs, e.g., in the review paper on PQC-based machine learning [Ref 1], and a few attempts preceded our work [20,21]. But, while these works did provide promising indications of learning, ultimately they did not succeed in achieving as good results in benchmarking environments as we did -- namely they did not achieve the (OpenAI gym-specified) threshold for considering the environments solved. Getting these levels required more than just a serious hyperparameter sweep. Our results only achieved after we introduced a few new original enhancements to standard PQCs; specifically the use of data re-uploading circuits with trainable input scaling parameters and trainable observables. This is one of the original features of our work that we believe is worthwhile to report.
>
> [Ref 1] Benedetti et al. Parameterized quantum circuits as machine learning models. Quantum Science and Technology (2019). https://iopscience.iop.org/article/10.1088/2058-9565/ab4eb5/meta

---

### Decision · Program_Chairs · 2021-09-27

**Decision:**

Accept (Poster)

**Comment:**

The authors present a quantum reinforcement learning regime that utilizes a parameterized quantum circuit. The paper is clearly written. There are a few concerns with the scientific contributions and side points from the reviewers, especially why the PQC plays an essential role in the learning model. However, the authors have replied explicitly the concerns in the discussion and the reviewers and the authors have reached consensus. The authors also explain in detail on possible applications of the work and the questions on the complexity analysis. In short, I would recommend this work to be accepted.